# P-selectin mobility undergoes a sol-gel transition as it diffuses from exocytosis sites into the cell membrane

Nicola Hellen[1], Gregory I. Mashanov[1], Ianina L. Conte [2], Sophie le Trionnaire [2], Victor Babich [3], Laura Knipe[1], Alamin Mohammed[2], Kazim Ogmen [2], Silvia Martin-Almedina[2], Katalin Török [2], Matthew J. Hannah[4], Justin E. Molloy [1✉] & Tom Carter [2✉]

In response to vascular damage, P-selectin molecules are secreted onto the surface of cells that line our blood vessels. They then serve as mechanical anchors to capture leucocytes from the blood stream. Here, we track individual P-selectin molecules released at the surface of live endothelial cells following stimulated secretion. We find P-selectin initially shows fast, unrestricted diffusion but within a few minutes, movement becomes increasingly restricted and ~50% of the molecules become completely immobile; a process similar to a sol-gel transition. We find removal of the extracellular C-type lectin domain (ΔCTLD) and/or intracellular cytoplasmic tail domain (ΔCT) has additive effects on diffusive motion while disruption of the adapter complex, AP2, or removal of cell-surface heparan sulphate restores mobility of full-length P-selectin close to that of ΔCT and ΔCTLD respectively. We have found P-selectin spreads rapidly from sites of exocytosis and evenly decorates the cell surface, but then becomes less mobile and better-suited to its mechanical anchoring function.

[1] The Francis Crick Institute, London, UK. [2] Molecular and Clinical Sciences Research Institute, St Georges University of London, London, UK. [3] Mercy College of Health Sciences, Des Moines, IA, USA. [4] Microbiology Services Colindale, Public Health England, London, UK. ✉email: justin.molloy@crick.ac.uk; tcarter@sgul.ac.uk

P-selectin (CD62P) is a type I transmembrane protein that is stored in specialized sub-cellular granules called Weibel–Palade Bodies (WPBs) that are present in blood vessel endothelial cells. It is secreted from the WPBs into the plasma membrane in response to vascular damage and functions in the early stages of defense against infection by recruiting circulating leukocytes from the bloodstream to the vessel wall. To capture leukocytes, P-selectin binds to a receptor, P-selectin glycoprotein ligand-1 (PSGL-1) present on the leukocyte membrane[1]. The P-selectin:PSGL-1 interaction has very fast binding ($k_{on}$) and fast unbinding ($k_{off}$) kinetics, thought to be important for efficient capture and subsequent rolling of leukocytes[2–4]. Bond lifetime (determined by $k_{off}$) is load-dependent and shows a combination of catch- and slip-bond behavior[5] and bound lifetime peaks at approximately ~10–20 pN force. Importantly, we do not understand how P-selectin can spread with ease across the plasma membrane of the endothelial cell and yet bear the high forces required for the catch-bond formation and efficient leukocyte capture. To address this, we used total internal reflection fluorescence (TIRF) video microscopy to study the movement of individual P-selectin molecules following stimulated release from WPBs in live human endothelial cells (HUVECs).

P-selectin is a modular protein consisting of a series of domains that determine its intracellular trafficking and its function on the cell surface (Supplementary Fig. 1). We reasoned that its structure was likely to impact its mobility at the plasma membrane. The extracellular portion consists of an N-terminal, $Ca^{2+}$-dependent (C-type) lectin domain (CTLD) responsible for binding to PSGL-1, but which can also bind glycosaminoglycans (GAGs)[6] found in the glycocalyx and basement membranes of endothelial cells[7]. Below the CTLD is an EGF-like domain and nine consensus repeat domains (CRs) that give an extended conformation of ~38 nm[8]. Removal of five or more CRs prevents leukocyte adhesion underflow, while disruption of the extracellular glycocalyx partially rescued the effect, indicating that the full-length CR region acts to project the CTLD through the surface glycocalyx[9]. Measurement of elasticity[10] and bending rigidity[11] suggest the extracellular portion of the molecule behaves as a linear spring capable of sustaining large forces. The transmembrane region consists of a single-pass helix which is followed by a short, intracellular, cytoplasmic tail (CT) that contains sorting information to direct newly-synthesized P-selectin to WPBs[12]. The WPB membrane, therefore, acts as a storage reservoir for P-selectin[13], so large amounts can be rapidly delivered to the plasma membrane following stimulated exocytosis. After stimulation, cell-surface P-selectin levels increase, up to fourfold, peaking by 2–3 min before declining to low levels over 10–20 min[14]. The in vivo estimate of the average P-selectin density following stimulation (25–50 molecules/μm²)[15] is similar to that required for leukocyte adhesion and rolling in vitro[16,17]. Clustering of P-selectin is seen at later times after secretion[16] that may arise through sequestration into clathrin-coated pits (CCPs)[18] and/or into special microdomains close to CCPs in a CD63-dependent fashion[19], although the precise mechanism remains unclear. Clustering may give rise to regions of high local density which may be important in preventing internalization and promoting cell-cell adhesion[18,19]. The CT does not bind actin or actin-binding proteins[20], but its interaction with other molecules, such as the AP2 complex[21] located at the plasma membrane may affect mobility.

In the current study, we use single fluorophore imaging to show how P-selectin molecules diffuse rapidly and in an unrestricted manner following release from sites of exocytosis, but over a period of several minutes, diffusive motion becomes anomalous and an increasing number of molecules become immobile. We present a simple mechanical model to describe the anomalous diffusive behavior and show that both intracellular and extracellular regions of P-selectin are important in progressively slowing and restricting P-selectin movement, making it better-suited to its known leukocyte anchoring function.

## Results and discussion

**Mobility of full-length P-selectin following exocytosis.** We first investigated the mobility of full-length P-selectin-eGFP by imaging the movement of individual molecules released by histamine-evoked WPB exocytosis. Regions of interest (ROIs) in video recordings, centered on individual WPBs, were concatenated so that multiple secretion events could be analyzed (Supplementary Movie 1). Single molecules (SMs) were identified by their characteristic diffraction-limited spot-size, single GFP intensity level, and single-step photobleaching behavior[22]. SMs were localized with ~25 nm precision in each video frame and tracked from frame to frame using image analysis software[23] (Fig. 1a–d). We validated our ability to track individual molecules present at different surface densities and with different diffusion coefficients using simulated data sets with identical signal-to-noise to our real data sets as described previously[24].

Molecular trajectories were analyzed by plotting mean-squared displacement against time interval (MSD vs $dT$). For unconstrained diffusion, trajectories should follow a random Brownian walk and MSD vs $dT$ plots will be linear with gradient $4D_{lat}$ (where $D_{lat}$ is the two-dimensional lateral diffusion coefficient). Deviation from linearity indicates anomalous diffusion; downward-curvature indicates constrained motion, upward curvature convective flow or motorized motion[22]. For the first few seconds after exocytosis MSD vs $dT$ plots were approximately linear, indicating unconstrained, diffusion (Fig. 1e). We found no evidence for motorized or convective motion. Each trajectory was color-coded according to its diffusion coefficient ($D_{lat}$), determined from the initial gradient of the MSD vs $dT$ plot, and all trajectories were then overlaid in a single image as a heat-map (Fig. 1a–d). The median $D_{lat}$ ~0.2 μm² s⁻¹ is similar to the value expected from the Saffman–Delbruck model for membrane protein diffusion[25]. We noticed that constitutively-released P-selectin molecules, present at low density prior to stimulation, showed highly-restricted, anomalous diffusion (Fig. 1a and Supplementary Fig. 2) which was in marked contrast to the freshly released P-selectin molecules (Fig. 1b). Based on this observation we explored how the diffusive motion of P-selectin changed with time after secretion. After 2–3 min, P-selectin had spread out over the entire plasma membrane (Fig. 1c), but the diffusive motion was now slower and more constrained (Fig. 1e) with MSD vs $dT$ plots showing distinct downward-curvature. Also, at later times, the number of immobile molecules increased significantly (Fig. 1d, f). These time-dependent changes in mobility appear similar to a sol-gel transition, like an agar solution gradually setting to a gel as it cools. Freshly released P-selectin molecules at first move freely but with time somehow become ensnared.

**Molecular basis for time-dependent changes in P-selectin mobility.** We compared a panel of P-selectin deletion mutants lacking specific functional domains, including the cytoplasmic tail (ΔCT), the C-type lectin domain (ΔCTLD), the CTLD and EGF-like domains (ΔCTLD-EGF), consensus repeat domains (Δ4CR and Δ8CR), and a highly-truncated molecule consisting of just the transmembrane helix, CR9 and EGF domains (which we term "TMD") (Supplementary Fig. 3a). All of the mutants with the exception of TMD localized correctly to WPBs (Supplementary Fig. 3b-h) and SMs were released during histamine-evoked

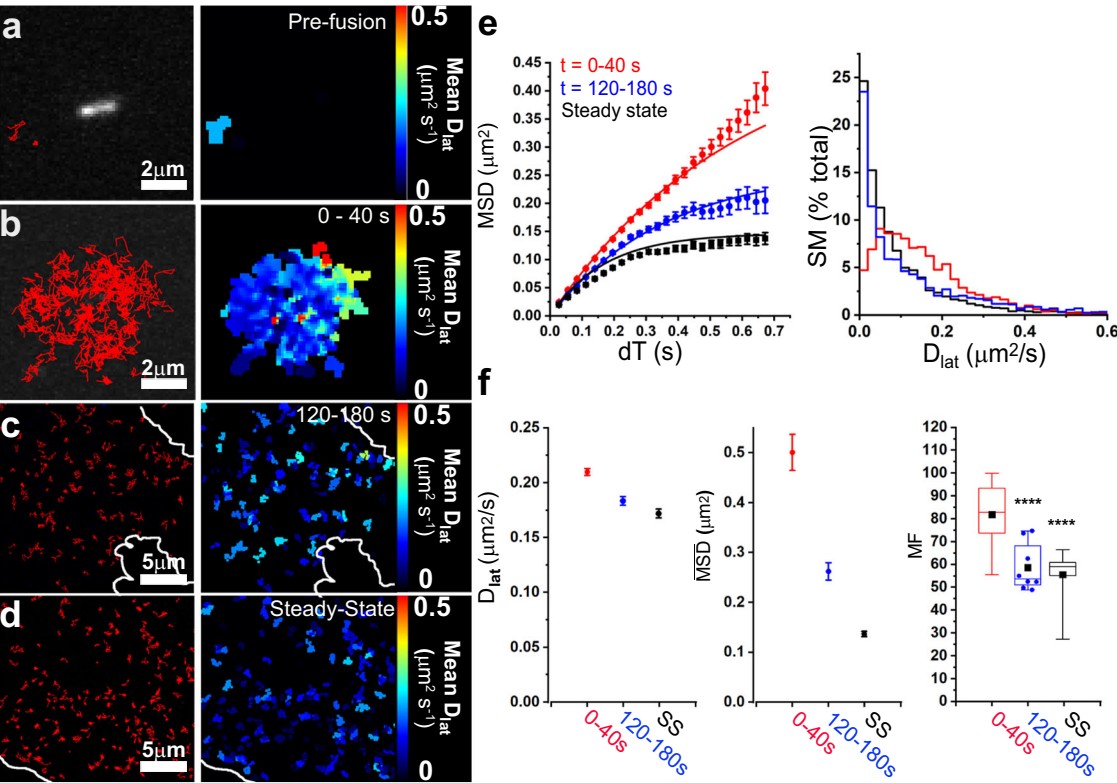

**Fig. 1 Mobility of full-length P-selectin-eGFP slows and becomes more restricted with time after exocytosis from WPBs. a–d** Single-molecule trajectories are shown for 40 s video records, before, during, and after exocytosis (left panels, red traces. The start time, $t = 0$, is defined as the moment that fusion occurs. **a** Molecules present at steady-state, recorded for 40 s prior to stimulation ($n = 2$ SMs). Note the Weibel–Palade Body present in the center; **b** Data recorded for 40 s immediately after exocytosis ($t = 0$–40 s) triggered by 100 μM histamine;. Here, the x-y centroids of five separate exocytosis events were set to the center of the image and videos were concatenated for analysis ($n = 129$ SMs, five fusion events); **c** $t = 120$–180 s after exocytosis and **d** under steady-state ("SS") conditions. In the corresponding panels (right) the same tracks are pseudo-colored to report their mean lateral diffusion coefficient ($D_{lat}$) over the track duration. White lines in **c**, **d** show cell boundaries. **e** MSD vs dT plots (left) (dots show the mean MSD value at each given dT and vertical bars delimit ± s.e.m.) and distributions of $D_{lat}$ (right); shown for different times after exocytosis: Red, $t = 0$–40 s (3522 SMs, $n = 66$ exocytosis events from 20 cells); Blue, $t = 120$–180 s post-exocytosis (2700 SMs, $n = 12$ cells) and Black: steady-state (3012 SMs, $n = 14$ cells). **f** $D_{lat}$ values and limiting MSD obtained by least-squares fitting the data in **e** (left) to Eq. (1) (vertical bars delimit ±95% confidence intervals of the least-squares fits); mobile fractions, MF, (defined as molecules with $D_{lat} > 0.05$ μm² s⁻¹): Shortly after exocytosis ($t = 0$–40 s) gave 82.0 ± 1.8%, s.e.m. ($n = 20$ cells, 66 fusion events, four experiments); post-exocytosis ($t = 120$–180 s) gave 57.4 ± 3.7% ($n = 8$ cells; three experiments); one-way ANOVA vs "0–40 s", ****$P = 0.0004$; steady-state ("SS") gave 54.3 ± 2.8% ($n = 14$ cells, four experiments); one-way ANOVA vs $t = 0$–40 s, ****$P < 0.0001$. Box and Whiskers: where: Boxes show 25–75% data range, horizontal line is median, solid symbol is mean, Whiskers delimit 5–95% of data, and all data points shown if $n < 10$.

exocytosis (Fig. 2a–c). Both ΔCTLD and ΔCTLD-EGF mutants showed an identical increase in mobility at exocytosis (Fig. 2b–d), indicating the N-terminal CTLD hinders P-selectin movement even at the earliest time point. The TMD mutant, which was present constitutively at the plasma membrane, exhibited rapid and unconstrained diffusion ($D_{lat} = 0.4$ μm² s⁻¹, see Fig. 3a) similar to ΔCTLD and ΔCTLD-EGF mutants at the time of exocytosis (Fig. 2b). The ΔCT, Δ4CR, and Δ8CR mutants showed little difference compared to the parent molecule at the time of secretion (summarized in Fig. 2b-d). At later times, the MSD vs dT relationships for ΔCTLD and ΔCT showed much less-pronounced downward-curvature than the full-length molecule (Fig. 3a, b). Together, these findings indicate that the time-dependent changes in mobility are unlikely to be due to lipid mixing between WPB and plasma membranes, but instead are controlled by the independent and additive effects of the pro-truding CTLD and CT domains.

**A simple mechanical model for changes in diffusive behavior.** The downward-curvature of MSD vs dT plots (Fig. 3a) is indicative of anomalous sub-diffusion which can arise from different physical phenomena; e.g., heterogeneity in lipid composition (e.g., lipid rafts) or interactions between P-selectin and components of cortical assemblies or the extracellular matrix leading to so-called "caged" or "hop-diffusion"[26]. Since the motion of proteins in lipid bilayers is over-damped, inertial forces can be ignored and the balance of thermal, viscous, and elastic forces can be modeled empirically as a Kelvin–Voigt mechanical system of parallel viscous ($\beta$) and elastic ($\kappa$) elements. The model, shown in Fig. 3c, gives an expected MSD vs dT relationship given by ref. [27] Eq. (1).

$$\text{MSD}_{\Delta t} = \frac{4 \cdot k_B T}{\kappa} \left( 1 - e^{\frac{-\kappa \Delta t}{\beta}} \right) \tag{1}$$

Where thermal energy is $k_B T$ and $\beta$ and $\kappa$ are the sum of viscous and elastic elements (Fig. 3e). The derivative of Eq. (1) gives an initial gradient (at $\Delta t = 0$): $\frac{4 \cdot k_B T}{\beta} = 4D_{lat}$ as expected for two-dimensional diffusion, while the limiting plateau amplitude ($\overline{\text{MSD}}$ at $\Delta t = \infty$) is given by: $\overline{\text{MSD}} = \frac{4 \cdot k_B T}{\kappa}$.

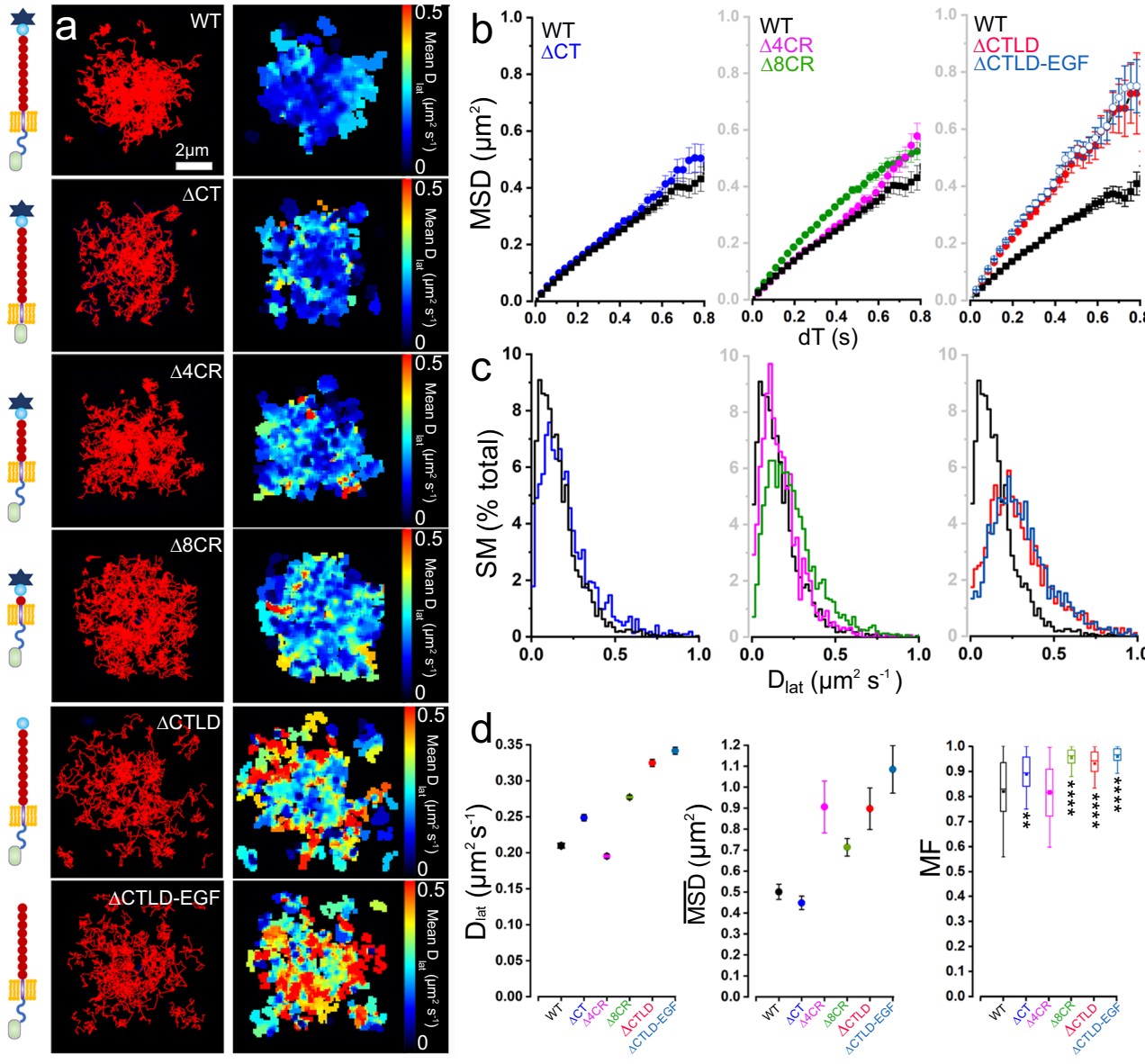

**Fig. 2 Deletion of CTLD increases P-selectin mobility at exocytosis sites. a** SM trajectories detected at WPB exocytosis sites immediately after histamine stimulation (left panels) shown for different deletion mutants. Video records were aligned and concatenated as described in Fig. 1b. From top: Full-length P-selectin-eGFP (WT) (165 SMs, $n = 9$ fusion events), $\Delta$CT (174 SMs $n = 5$ events), $\Delta$4CR (265 SMs, $n = 5$ events), $\Delta$8CR (187 SMs, $n = 5$ events), $\Delta$CTLD (170 SMs, $n = 4$ events), and $\Delta$CTLD-EGF (146 SMs, $n = 3$ events). Scale bar applies to all images. Tracks are pseudo-colored to report mean lateral diffusion coefficient ($D_{lat}$) over track duration (right panels). **b** MSD vs dT plots (as for Fig. 1e) for mutants compared to WT (black data, replicated in the plots to ease comparison). **c** $D_{lat}$ distributions for mutants compared to WT (black data). For **b**, **c** $\Delta$CT (blue; 2244 SMs, $n = 41$ fusion events, $n = 15$ cells, three experiments); $\Delta$4CR (magenta; 1502 SMs, 72 fusion events, $n = 28$ cells, three experiments); $\Delta$8CR (green; 3907 SMs, 50 fusion events, $n = 19$ cells, three experiments); $\Delta$CTLD (red; 872 SMs, 57 fusion events, $n = 12$ cells, three experiments) and $\Delta$CTLD-EGF (dark blue; 1569 SMs, 37 fusion events, $n = 8$ cells, three experiments). **d** Fitted values of $D_{lat}$ ($\pm 95\%$ confidence interval) and limiting MSD ($\overline{MSD}$) ($\pm 95\%$ confidence interval). Mobile fraction, MF, (defined as molecules with $D_{lat} > 0.05\ \mu m^2\ s^{-1}$) plotted as Box and Whiskers (as defined in Fig. 1). Results of one-way ANOVA, multiple comparison, Dunnett tests indicated as: "*****"$P < 0.0001$, "**"$P = 0.0035$, no stars not significant.

MSD vs $dT$ plots were fitted individually to Eq. 1 to give an estimate of the lumped $\kappa$ and $\beta$ for each protein tested and these values were then used to derive the diffusion coefficient, $D_{lat}$ and $\overline{MSD}$ (Fig. 3d). Global fitting (Microsoft Excel "Solver" function) to all of the MSD vs $dT$ data sets by variance-weighted, least-squares minimization, using the equations given in Fig. 3e (line fits shown in Fig. 3a) gave estimates of $\kappa_{CT} = 45\ nN\ m^{-1}$; $\kappa_{CTLD} = 65\ nN\ m^{-1}$ and $D_{lat} = (k_BT/\beta_{CT}) = 0.48\ \mu m^2\ s^{-1}$; $(k_BT/\beta_{TMD}) = 0.41\ \mu m^2\ s^{-1}$; $(k_BT/\beta_{CTLD}) > 10\ \mu m^2\ s^{-1}$. The results are consistent with a simple visual inspection of the graphs which

indicates the asymptotic diffusion distance, $\overline{MSD}$, for the full-length protein (given by $4k_bT/(\kappa_{CTLD} + \kappa_{CT}) \approx 0.15\ \mu m^2$) increases by a similar amount (to $\sim 0.35\ \mu m^2$) when either the CT or CTLD regions are removed and when both regions are removed (TMD mutant) the $\overline{MSD}$, becomes larger still (in our simple model tending to infinity). So, the simplest interpretation is that the CT and CTLD regions act as independent visco-elastic elements that are combined in parallel with viscous drag contributions dominated by the membrane-spanning helix (TMD) and CT regions (Fig. 3c).

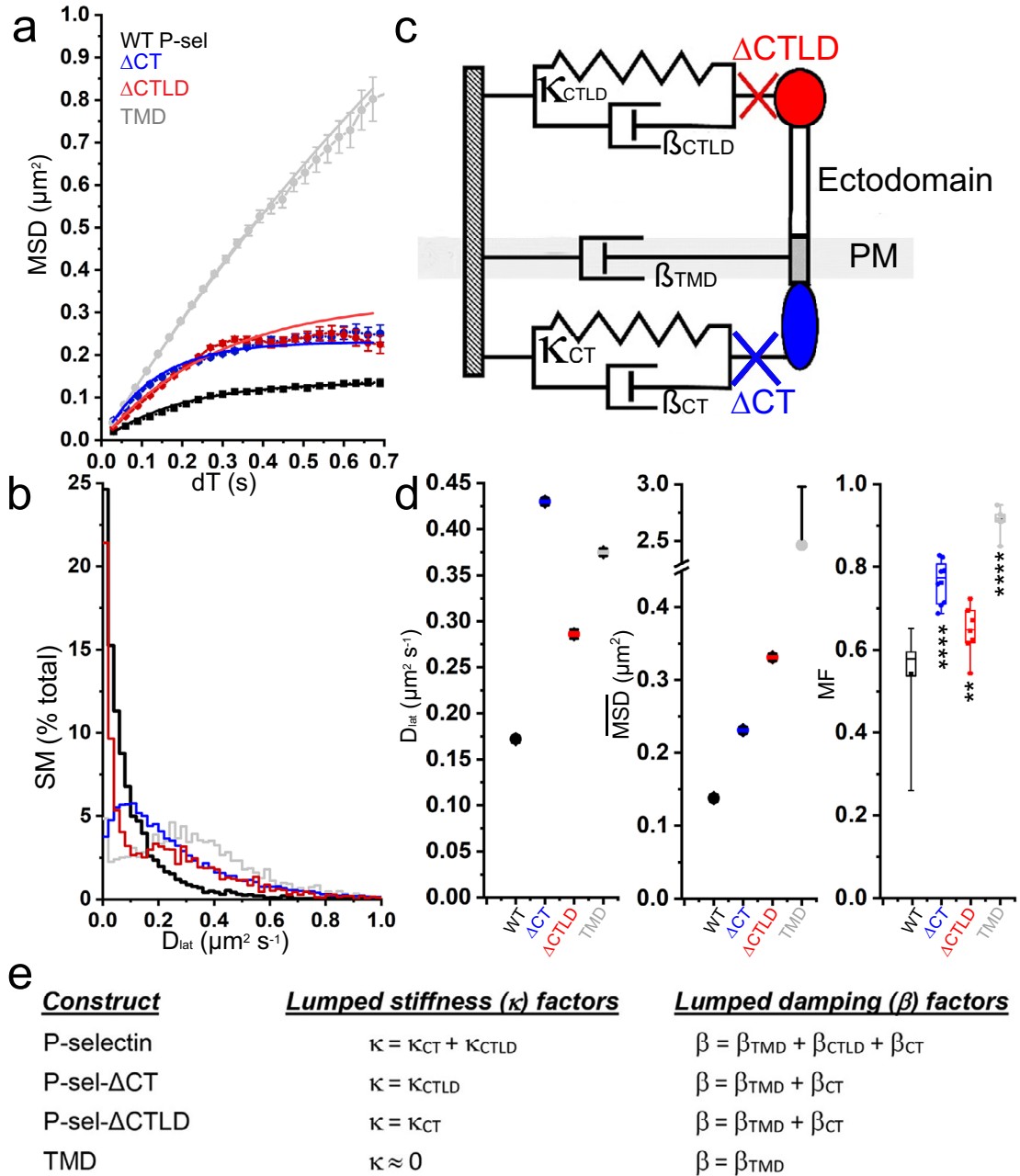

**Fig. 3 CT and CTLD independently shape P-selectin mobility at steady-state. a** Plots of *MSD* vs *dT* for ΔCTLD (Blue; 2621 SMs, *n* = 6 cells), ΔCT (Red; 6080 SMs, *n* = 8 cells) and TMD (Light Gray; 1687 SMs, *n* = 6 cells) under steady-state conditions. For comparison, data for full-length P-selectin (from Fig. 1e) is shown in Black. Solid lines are least-squares fits by global fitting (see main text and panels **c**, **e**). **b** $D_{lat}$ distribution for each molecule, color-coded as in **a**. **c** Mechanical model showing how intracellular, transmembrane and extracellular regions of P-selectin may act as independent Kelvin–Voigt elements which in different combinations give the MSD vs dT fits shown in **a**. CT (red) and CTLD (blue) regions contribute visco-elasticity while the TMD (gray), acts only as a viscous element. Red and blue crosses indicate how ΔCT and ΔCTLD mutations change the mechanical coupling. **d** $D_{lat}$ (fitted values ±95% confidence intervals), limiting amplitude ($\overline{MSD}$) (fitted values ±95% confidence intervals), and MF (Box and Whiskers as defined in Fig. 1) for the different protein constructs (as indicated). Results of one-way ANOVA, multiple comparisons, Dunnett tests indicated as: "****"$P < 0.0001$; "**"$P = 0.008$, **e** Shows how the Kelvin–Voigt elements (shown in **c**) combine in different ways to give the lumped stiffness, $\kappa$ and damping factor, $\beta$ used for global fitting (shown in panel **a**) to the WT and truncation mutant data.

**High-resolution single fluorophore tracking reveals stochastic pausing**. To gain a better understanding of the physical factors that might manifest as springs and dashpots in the model, thereby shaping the mobility of full-length P-selectin, we used the bright and photostable fluorophore, Cy3B, to perform high-precision SM localization over extended time periods. The P-selectin ectodomain was labeled using the anti-P-selectin antibody, AK6, covalently-conjugated to Cy3B (AK6-Cy3B), and as control Cy3B labeled

isotype mouse IgG (IgG-Cy3B) using conditions that gave ~1:1 labeling stoichiometry (see Methods and Supplementary Fig. 4). AK6 is thought to bind within the CR region of P-selectin's ectodomain and was chosen because it does not interfere with P-selectin function[28] or the internalization and intracellular trafficking of the molecule[12]. Binding specificity in live cells was confirmed using IgG-Cy3B as a negative control (Supplementary Figs. 4h-k and 5a). SM tracking of AK6-Cy3B bound to P-selectin gave MSD vs *dT*

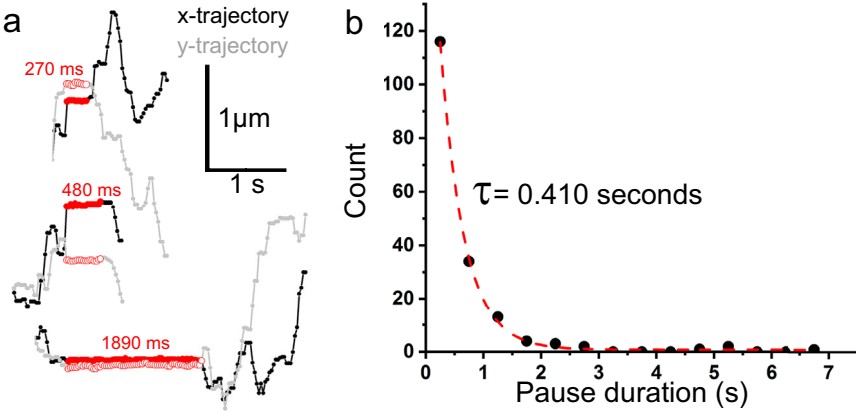

**Fig. 4 P-selectin undergoes transient binding interactions during diffusion at steady-state. a** Examples of x-y trajectories for AK6-Cy3B-labeled P-selectin under steady-state conditions. Regions marked in red, highlight brief pauses in motion. **b** Distribution of pause durations and lifetime, $\tau$ (176 events, bin width 0.5 s).

relationships, estimates of $D_{lat}$ and MF similar to full-length P-selectin-eGFP (Supplementary Fig. 5c, d). Two-color imaging confirmed single AK6-Cy3B and P-selectin-eGFP co-localized and moved together (Supplementary Fig. 5e). Using AK6-Cy3B we confirmed that the steady-state mobility of P-selectin on the apical membrane was identical to that on the basolateral membrane (Supplementary Fig. 6). Close inspection of the AK6-Cy3B data revealed that a small number of trajectories (3.4%) showed diffusive motion that was interspersed with distinct pauses (Fig. 4a). Pause durations showed an exponential distribution with a time constant of ~0.4 s (Fig. 4b). The stochastic stalling behavior may belie the cause of restricted anomalous diffusion observed for full-length P-selectin at steady-state.

**Identifying molecular interactions that limit P-selectin mobility.** To further investigate the molecular basis for the independent contribution of the CT to P-selectin mobility we first disrupted the actin cytoskeleton using cytochalasin D. We found little effect on steady-state P-selectin mobility (Supplementary Fig. 7) consistent with previous studies indicating P-selectin does not bind actin or actin-binding proteins[20]. The best-characterized plasma membrane-associated interactor for P-selectin CT is the μ2 subunit of AP2[21]. Binding to AP2 is thought to cluster P-selectin at the cell surface, a process that may facilitate its function as a leukocyte adhesion molecule[18]. In live cells, we observed a population of immobile eGFP puncta with intensities that were greater than expected for individual fluorophores. A histogram of the intensity distribution measured in the eGFP channel for these puncta showed a broad distribution with an average value equivalent to ~8 eGFP molecules suggesting they reflect regions of P-selectin clustering (Supplementary Fig. 8). Further analysis showed that a subset of P-selectin-eGFP puncta co-localize with endogenous AP2 at the cell surface in a CT-dependent fashion (Supplementary Fig. 9). To test the role of AP2 in shaping P-selectin mobility we disrupted the AP2 complex by siRNA (Supplementary Fig. 10), and found that both the MF and $\overline{MSD}$ for P-selectin-eGFP significantly increased at steady-state (Fig. 5). A similar change was observed with the ΔCTLD mutant (Fig. 5). We next over-expressed AP2μ2-mCherry and confirmed that it co-localized with the endogenous AP2 complex and P-selectin-eGFP in a CT-dependent fashion (Supplementary Fig. 11). Overexpression of AP2μ2-mCherry was found to reduce $\overline{MSD}$ but not mobility (Fig. 5b, c). The effects of either knocking down or over-expressing AP2 are consistent with its interactions with the CT region of P-selectin causing reduced mobility at

steady-state. We noticed that disruption of the AP2 complex did not render the ΔCTLD mutant freely mobile (i.e., like TMD) and this may reflect the presence of residual AP2 in the siRNA-treated cells, although additional mechanisms may operate to restrict P-selectin diffusion. It is known that CD63 plays a role in leukocyte adhesion[19,29] and that it co-localizes with P-selectin in clusters distinct from CCPs on the plasma membrane[19]. In preliminary dual-color single fluorophore tracking experiments, we found that although the majority of individual eGFP-CD63 and Cy3B-labeled P-selectin molecules diffused independently, there was a small proportion (~0.5%) of freely-diffusing co-incident single-molecule tracks while about 5% of the total signal co-localized to immobile puncta which exhibited fluorescence greater than expected for a single eGFP:Cy3B fluorophore pair. Our estimates of CD63:P-selectin complex formation should be considered lower bounds because of the presence of unlabeled CD63 and P-selectin in the cells. Importantly, these experiments raise the possibility that some mobile P-selectin molecules might be chaperoned by CD63 while others[19], might form static, multimeric, complexes with CD63 (Supplementary Fig. 12).

We next looked at the effects of the CTLD on P-selectin mobility. It is known that the CTLD can bind heparan sulfate (HS), the major extracellular glycan species[6] present in the apical glycocalyx and basement membrane of endothelial cells[7]. So, we hypothesized that the binding of CTLD to HS might also shape the diffusive behavior of P-selectin. To test this, we removed cell-surface HS using heparanase III (Fig. 6a, b) and found an increase in $D_{lat}$ and MF for P-selectin-eGFP to levels similar to ΔCTLD (Fig. 6g, h). Analysis of individual trajectories using AK6-Cy3B labeling showed a tenfold reduction in the frequency of pausing events following heparanase III treatment, (0.45%, $n = 5431$ molecular trajectories, six cells), although pause durations were similar ($\tau = 0.45$ vs 0.46 s, respectively). Treatment of cells to remove chondroitin sulfate (chondroitinase ABC, Fig. 6c, d) or sialic acid (neuraminidase; Fig. 6e, f) did not alter P-selectin mobility (Fig. 6g, h). Finally, to test whether P-selectin mobility is altered in cells exposed to pro-inflammatory stimuli, we treated HUVEC with IL-1ß under conditions that upregulate cytokine expression[30] and found no significant change in steady-state mobility of full-length P-selectin (Supplementary Fig. 13).

It is possible that the mixing of WPB membrane lipids with the plasma membrane at the time of exocytosis might generate a lipid microenvironment with distinct properties[31]. However, lipid released from the WPB envelope should mix and equilibrate rapidly with the plasma membrane lipids and the fast and unconstrained motion of the TMD mutant seen at steady-state,

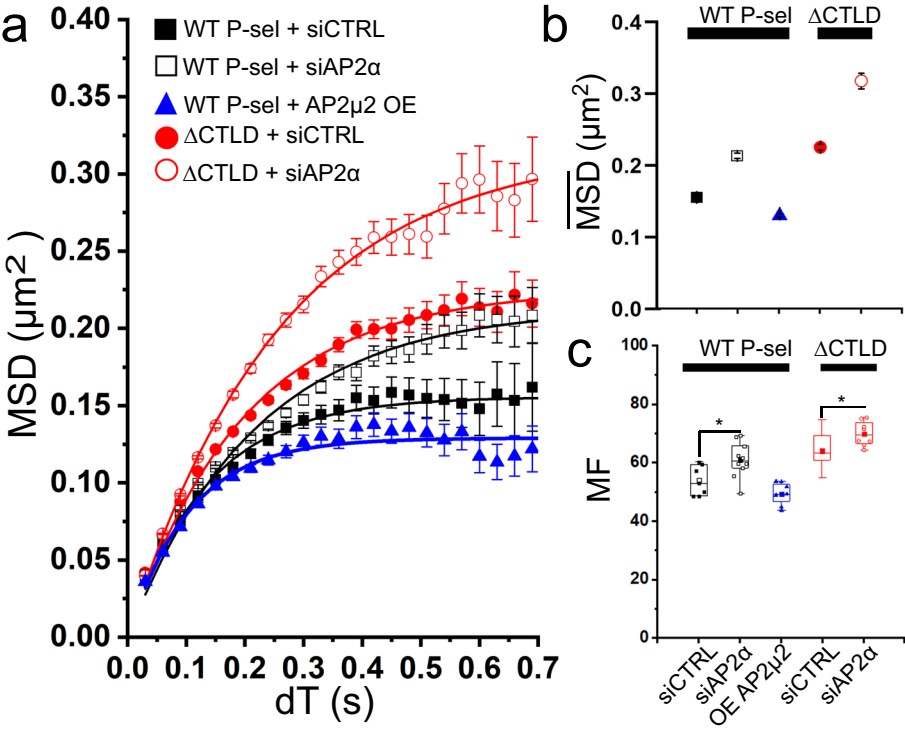

**Fig. 5 At steady-state P-selectin-CT shapes mobility via AP2. a** MSD vs *dT* plots show: WT siControl (solid back squares; 5056 SMs, *n* = 7 cells, two experiments) compared to WT with siAP2α treatment (open black squares; 9307 SMs, *n* = 14 cells, two experiments) and WT with AP2μ2-mcherry overexpression ("OE") (solid blue triangles; 5629 SMs, *n* = 7 cells, two experiments). The effect of siAP2α treatment on ΔCTLD mobility (open Red circles; 8300 SMs *n* = 17 cells, two experiments) is compared with ΔCTLD siControl (solid red circles; 19,391 SMs, *n* = 18 cells, two experiments). Data were collected under steady-state conditions. Data points are means, error bars ± s.e.m., solid lines are non-linear least-squares fits Eq. 1. **b** Limiting amplitude for diffusion ($\overline{\text{MSD}}$) (fit value ±95% confidence intervals) and **c** the MF plotted as Box and Whiskers as defined in Fig. 1. Results of one-way ANOVA, multiple comparisons, and Tukey test, indicated as: "*"*P* = 0.04.

strongly suggests that changes in the lipid phase (reported by $\beta_{\text{TMD}}$ in our model) cannot account for the time-dependent changes in the anomalous diffusion of P-selectin following exocytosis. Instead, our results show that the independent and additive interactions between the CT and CTLD regions with intra- and extracellular assemblies are the dominant contributors to the time-dependent restriction of P-selectin mobility. At steady-state we observed P-selectin-eGFP clustering into immobile puncta, that localized with the CCP-associated component AP2 (Supplementary Figs. 8 and 9). Clustering of secreted P-selectin into immobile structures may provide a way of increasing the local concentration of molecules thereby enhancing avidity by increasing the number of CTLD-PSGL-1 bonds providing a more stable platform to support leukocyte adhesion[18,19].

Restriction to mobility contributed by the ectodomain was mediated primarily by the CTLD region, whereas the length of the molecule (determined by the number of CR domains) played only a minor role, perhaps by positioning the CTLD within the extracellular environment. The restriction to mobility imposed by the CTLD region both at the time of exocytosis and later at steady-state, suggests CTLD interacts with the extracellular glycocalyx soon after secretion. It has been shown previously that the mobility of integral membrane proteins can be modified through extracellular interactions of their ectodomains[32,33]. The apical and basolateral membranes of endothelial cells are covered by a complex assortment of membrane-bound or associated proteo- and lipido-glycans[34]. HS is the major glycosaminoglycan (GAG; 50–90%) on the endothelial cell-surface[35] and plays a vital role in inflammation by presenting chemokines and growth factors to adherent leukocytes as they roll and sample the endothelial surface[34]. The CTLD region binds HS[6], and our observation that removal of HS

(but not other extracellular GAGs) increased the mobility of full-length P-selectin close to that seen with ΔCTLD mutant provides strong evidence that this GAG shapes P-selectin mobility via interaction with CTLD. Brief pauses in P-selectin movement, consistent with transient binding events (Fig. 4), were dramatically reduced in frequency after HS removal, consistent with CTLD-HS interactions causing stalling. Our finding that HS removal enhances P-selectin mobility and reduces its anomalous diffusive behavior, is counter to the recent observation of enhanced diffusion through polymer gels[36]. However, such gel network models (with different assumptions or working parameters) may provide a useful basis for better understanding our results.

In conclusion, we have found that time-dependent changes in mobility of P-selectin are controlled by both the CTLD and CT regions of the molecule. While the CTLD region impacts mobility via its interactions with cell-surface heparan sulfate, the CT contribution is relieved by knock-down and enhanced by overexpression of AP2. At steady-state, both regions, act independently and in parallel (Fig. 3c) tending to restrict mobility and thereby anchor P-selectin making it better-suited to its role in leukocyte capture. The relatively slow timescale of changes in mobility (i.e., *t* = 3 min) is important because it permits P-selectin to diffuse a reasonable linear distance, $\sqrt{2D_{\text{lat}}t} \sim 10\mu m$, from the exocytosis site and spread out across the membrane before it becomes less mobile. We still do not understand the detailed chemical changes responsible for P-selectin immobilization.

## Methods

**Cell culture, transfection, and immunocytochemistry.** Human pooled, primary Umbilical Vein Endothelial Cells (HUVEC) (catalog C12203, PromoCell GmbH, Heidelberg, Germany) were cultured (maximum passage, P4) in Medium 199

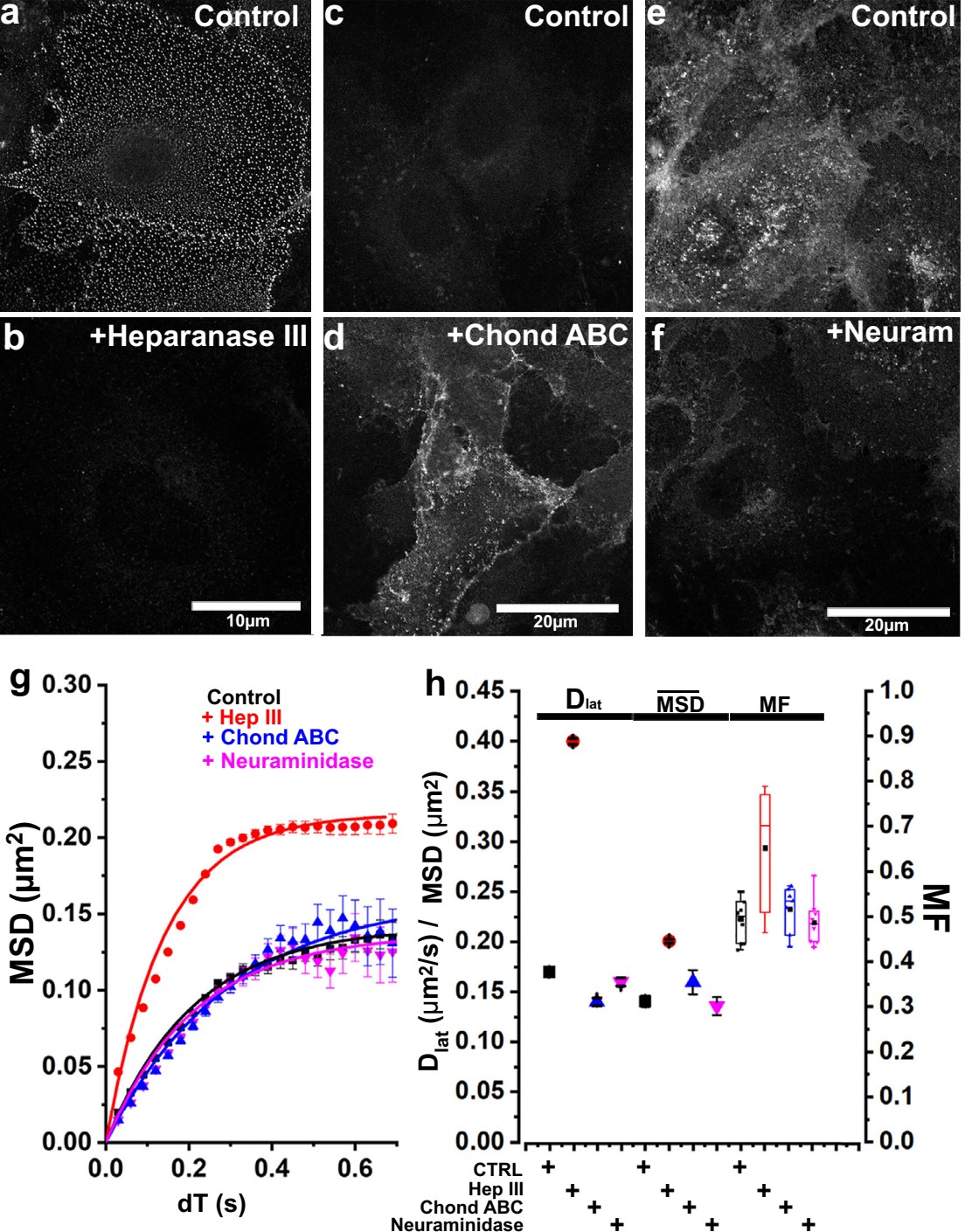

**Fig. 6 Plasma membrane heparan sulfate slows P-selectin mobility. a** Confocal images of fixed HUVEC immuno-labeled with heparan sulfate-specific antibody. **b** After treatment with heparanase III. **c** Confocal images of fixed HUVEC immuno-labeled with an anti-chondroitin 4 sulfate-specific antibody. **d** After treatment with chondroitinase ABC. Note that the antibody used here recognizes the digested chondroitin sulfate. **e** Confocal images of fixed HUVEC labeled with FITC-wheat germ agglutinin. **f** After treatment with neuraminidase. The images shown are representative of three independent experiments for each treatment. **g** MSD vs $dT$ plots for full-length P-selectin mobility measured in live cells before (black squares, 5918 SM, $n = 10$ cells) and after heparanase III treatment (red circles, 20,098 SM, $n = 21$ cells), chondroitinase ABC (blue triangles, 1482 objects, $n = 7$ cells), and neuraminidase (magenta triangles, 1044 objects, $n = 8$ cells). Solid lines are fits to the Kelvin–Voigt model. **h** Changes in mean $D_{lat}$ and limiting $MSD$ ($\overline{MSD}$) derived by least-squares fitting to the Kelvin–Voigt model (solid lines shown in panel **g**) error bars show 95% confidence intervals. Mobile fraction (MF) shown as Box and Whiskers as described in Fig. 1. One-way ANOVA multiple comparisons, Dunnett tests indicated chondroitinase ABC and neuraminidase treatments had no significant effect whereas Heparanase III treatment gave $P = <0.0004$ compared to control.

Earle's salts + L-Glutamine (Catalog 11150059, Thermo Fisher), supplemented with 20% fetal calf serum, 30 µg/ml Endothelial Cell Growth Supplement, 10 U/ml heparin and 50 µg/ml gentamicin at 37 °C in a 5% $CO_2$ atmosphere. Transfection was by a Nucleofector 2b device (catalog AAB-1001, Lonza, Basel, Switzerland) according to manufacturers' instructions[37–39]. Dharmacon ON-TARGET-plus siControl oligonucleotides and pre-validated ON-TARGET-plus siAP2α (siAP2α2) specific oligonucleotide (AAGAGCATGTGCACGCTGGCCA, Horizon Discovery Bioscience, UK[40]) with custom UU overhangs were used. Cells were transfected using 200 pMol siRNA, cultured for 48 h before the second round of transfection with 200 pMol siRNA and P-selectin-eGFP or mutant and 24 h later either fixed for immunocytochemistry or used for live-cell TIRFM imaging.

To remove surface glycans, cultured cells were treated with Heparanase III (3 U/ml, 1 h), Chondroitinase ABC (1 U/ml, 1 h) or Neuraminidase (0.1 U/ml, 1 h) (all reagents from Merck Life Science UK Limited, Gillingham, UK). Antibodies used for immunocytochemistry are in Supplementary Table 1. For immunolabelling, HUVECs were grown on porcine skin gelatin-coated 9 mm diameter, glass coverslips (Catalog G1890, Sigma), fixed with 3% paraformaldehyde in PBS supplemented with 0.2% (w/v) gelatin, 0.02% (w/v) saponin, 3 mM $NaN_3$, for 15 min prior to antibody labeling. Coverslips were mounted in Mowiol mounting media (Merck, Gillingham, Dorset, UK) and imaged 24 h later.

**AK6-Cy3B conjugation and P-selectin labeling**. For Cy3B conjugation to the anti-P-selectin Ab, AK6, or control IgG, the Amersham $CY^{TM}3B$ mono-reactive NHS ester dye labeling kit was used (GE Healthcare, Buckinghamshire, UK). About 1 mg of dye was dissolved in 100 µl of anhydrous DMSO and the dye concentration was determined by NanoDrop 1000 version 3.6.0 (Thermo Fisher Scientific) using a molar extinction coefficient for Cy3B of 130,000 $M^{-1} cm^{-1}$ at 563 nm, after adjustment for the reported batch purities for the reactive NHS ester (lot number 9674850; 82.3%, lot number: 9762352; 86.0%). Mouse IgG (AK6 or control IgG) concentration was calculated using the NanoDrop's preinstalled settings (210,000 $M^{-1} cm^{-1}$ at 280 nm) and the Dye:Protein (Cy3B:IgG) ratio after conjugation, was determined using dual absorbance measurements for IgG and Cy3B. The conjugation reaction reactive dye:IgG molar ratio required to give an approximate 1:1 labeling stoichiometry was determined empirically as 7:1. The reaction was carried out for 60 min at pH 8.5 using sodium borate buffer in the dark and stopped by the addition of an equal volume of sodium dihydrogen orthophosphate (pH 6, 0.2 M). Dye-protein conjugates were separated from unconjugated Cy3B NHS ester using a 7000 molecular weight cut-off Zeba$^{TM}$ Spin gel filtration desalting spin column (Thermo Scientific, Waltham, MA, United States) equilibrate with 2.5 mL of PBS. About 100 µl ultrapure water was added as a stacker prior to the addition of the reaction mixture before centrifuging at 16,100 × g for between ~12 min using an Eppendorf centrifuge (5415 D). Dye-protein conjugates were stored in the dark at 4 °C. For analysis of P-selectin labeling, internalization, and trafficking in live cells, AK6-Cy3B or control IgG-Cy3B (1:10 dilution) was added to live HUVEC 4 h post nucleofection with WT P-selectin-eGFP and then incubated for 24 h prior to fixation and processing for immuno-labeling with anti-GFP (secondary Cy2) and endogenous VWF (secondary Cy5). For labeling fixed cells with AK6-Cy3B or IgG-Cy3B (1:25), the conjugates were added alone or in combination with primary antibodies during the labeling protocol. For live-cell tracking experiments non-transfected, P-selectin-eGFP or eGFP-CD63 transfected HUVEC (24 h post-transfection) were incubated with AK6-Cy3B or control IgG-Cy3B (1:10 dilution) as appropriate, for 30 min at room temperature in the dark, before being carefully washed prior to imaging.

**DNA constructs**. WT P-selectin-eGFP, P-selectin-ΔCT-eGFP, and P-selectin-Δ8CR-eGFP were made as previously described[13,41]. P-selectin-ΔCTLD-eGFP, P-selectin-ΔCTLD-EGF-eGFP, P-selectin-Δ4CR-eGFP (4CRs removed), and P-selectin-Δ8CR-eGFP (8CRs removed), were made by inverted PCR from the WT P-selectin–eGFP plasmid, utilizing primers with EcoRV site introduced for self-ligation. Primers used for each mutant are shown in Supplementary Table 2. The P-selectin-ΔCT–eGFP construct was generated by removal of the cytoplasmic tail (exons 15 and 16) using primers 5′-CAATTCATCTGTGACGAGGG-3′ (forward) and 5′-G CAAAGTCTGTTTTTCTACTACCCAGCTGATA-3′ (reverse, introducing SalI site). The PCR fragment was digested using PvuI and SalI and ligated into PvuI–SalI-digested P-selectin–eGFP. A different strategy was used to generate P-Selectin-ΔCT-8CR-eGFP; the P-selectin-Δ8CR-eGFP was digested with XcmI and XhoI restriction enzymes and the fragment obtained inserted into XhoI-XcmI digested P-selectin-ΔCT–eGFP. A similar strategy was employed to generate P-Selectin-TMD-eGFP; P-Selectin-ΔCT-eGFP was digested with XcmI and AgeI restriction enzymes and a short fragment was ligated into XcmI-AgeI digested P-Selectin-ΔCT-8CR-eGFP. All constructs were sequence verified. AP2u2-mCherry[42] was a gift from Christien Merrifield (Addgene plasmid # 27672; http://n2t.net/addgene:27672; RRID:Addgene_27672). Clathrin-GFP[43] was from Prof. Margret Robinson (Cambridge University). eGFP-CD63 was a gift from Prof. Paul Luzio, Cambridge University, UK.

**Fluorescence microscopy**. Confocal microscopy was performed using a Leica TCS SP2 or Bio-Rad Radiance 2100 confocal microscope[39,44] equipped with PL APO 100 × 1.4 NA objective (SP2) or Nikon 60 × 1.4 NA objective lens (Bio-Rad). SM

detection and tracking experiments were carried out using a custom-made objective-based Total Internal Reflection Fluorescence (TIRF) microscope[22,45]. Dual-color imaging of eGFP and Cy3B was performed using 488 and 561 nm lasers (LightHUB, Omicron, Germany). The two fluorescence emission bands were separated using a dichroic mirror (FF552-Di02) angled at 45° to the incident light path and the two resulting, orthogonal light paths were directed via bandpass filters (FF01-525/50 and FF01-593/40, Semrock, NY) onto two separate EMCCD cameras (iXon897BV, Andor Technology Ltd., Belfast, UK). The pixel positions and image focus for the two cameras were co-aligned using a 3-axis positioning device and multi-colored, 100 nm diameter, fluorescent beads adhered to a microscope coverslip were used as fiducial markers (TetraSpeck, Thermo Fisher). Video data were acquired at either 30 or 40 frames s$^{-1}$ and analyzed using freeware custom software ("GMimPro" available at www.mashanov.uk). Analysis of single AK6-Cy3B fluorophore dwell times was carried out as previously described[46]. The locations of individual eGFP or Cy3B fluorophores were tracked automatically with sub-pixel (~20 nm) resolution. The software tracking criteria were set so that objects had fluorescence intensity similar to a single fluorophore, a waist diameter of ~250 nm, movement <5 pixels (~500 nm) between adjacent frames, and a continuous track length spanning >15 frames. Data for several thousand individual molecules were obtained from multiple exocytosis events and multiple cells by pooling data sets so that MSD vs dT plots, histograms of $D_{lat}$ estimates, and the immobile fraction for each experimental condition could be quantified. The mobile fraction (MF) was defined by the fraction of SMs with $D_{lat} > 0.05 \mu m^2 s^{-1}$. Cells were imaged 24–48 h after nucleofection. For analysis of steady-state mobilities, cells were imaged 24 h after nucleofection and selected on the basis of there being abundant single molecules at the plasma membrane but few or no WPBs. For analysis of mobilities during (0–40 s) and shortly after (120–180 s) exocytosis, cells were imaged 48 h after nucleofection and selected on the basis of well-spaced WPBs and few detectable molecules at the plasma membrane.

**Statistical analysis**. We carried out between three and nine replicate experiments for WT P-selectin and each of the various mutants. For the RNAi and AP2μ2 OE experiments, we carried out three and two replicates respectively. Data in each of the plots were pooled from three (or in the case of OE n = 2) replicate experiments, and are representative of the data from other replicate experiments. Data were plotted in Origin 2018 or GraphPad Prism 8.0.2. Linear and non-linear (exponential) fitting was carried out in Origin 2018 (OriginLabs, Northampton, MA), IGORPRO (WaveMetrics, Portland OR), or Microsoft Excel solver function. Statistical analysis was done by t-test (nonparametric), or by one-way ANOVA using Tukey or Dunnett's multiple comparisons test as indicated (GraphPad Prism 8.0.2). Significance values (P) for t-test or adjusted P values for ANOVA comparisons are shown where appropriate. MSD vs dT plots were constructed by computing MSD for all pairwise dT intervals for every trajectory in the data set. The MSD estimate at each dT was plotted with s.e.m. shown as error bars. Estimates of $D_{lat}$ and $\overline{MSD}$ derived by non-linear least-squares fitting to the MSD vs dT plots are displayed as best fit value ±95% confidence interval. Mobile fractions are plotted as Box and Whisker diagrams (as described in the Figure legends).

**Reporting summary**. Further information on research design is available in the Nature Research Reporting Summary linked to this article.

## Data availability

Source Data is provided in MS Excel format accompanying this manuscript available online at Nature Communications. An example video, Supplementary Movie 1 is provided online. We find compressed video formats (e.g., AVI, MP4, MOV, WMV, FLV, etc) of mobile single molecules are of low quality as compression algorithms perform poorly with this type of video data. Raw video imaging data, consisting of many tens of gigabytes of separate video data files, are available from the corresponding authors upon reasonable request.

## Code availability

The software used for video image analysis and simulation of single fluorophore dynamics is free to download (GMimPro 64-bit Ver 2022. http://www.mashanov.uk/) and used as described in earlier publications[23,24]. Other software is either commercial or freeware: Origin 2018 sr1 (OriginLabs, One Roundhouse Plaza, Northampton, MA, USA); GraphPad Prism 9.0.2 (Northside Dr. San Diego, CA, USA); Adobe Photoshop CC vr 23.2.2; ImageJ ver1.53f, http://imagej.nih.gov/ij; MS Office Professional Plus 2019 with Solver add-in, GRG Non-linear, least-squares minimization (Frontline Systems, Inc. Incline Village, NV. USA); IgorPro 8.0, Wavemetrics (Lake Oswego, OR, USA).

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

## Acknowledgements

UK MRC program grant: MC_PC_13053 (T.C.), Francis Crick Institute core funding from Cancer Research UK: FC001119 (JEM), UK Medical Research Council: FC001119 (J.E.M.), Wellcome Trust: FC001119 (J.E.M.), BBSRC grant: BB/S003894/1 (K.T. and T.C.). We thank Ms. Ayoni Medagoda and Mr. Acieb Karasi for help with Cy3B labeling of AK6 and associated immunocytochemistry.

## Author contributions

N.H., V.B., M.J.H., G.I.M, J.E.M., A.M., and K.T. generated vital reagents, software, or equipment. N.H., G.I.M., T.C., I.L.C., S.L.T., L.K. S.M.-A., and K.O., performed research. N.H., G.I.M, S.L.T., T.C., and J.E.M., analyzed data; J.E.M and T.C. designed the research and wrote the paper. The authors report no disclosures.

## Funding

## Competing interests

The authors declare no competing interests.
