## [Peer Review File · Nature Communications]

P-selectin undergoes a sol-gel transition as it diffuses from exocytosis sites into the endothelial cell plasma membrane.REVIEWER COMMENTS

Reviewer #1 (Remarks to the Author):

The manuscript of Hellen et. al., examined the trajectory of P-selectin secreted from Weibel Palade bodies following histamine stimulation. This study lacked a stated hypothesis to guide the major question of how P-selectin diffuses across the plasma membrane of activated endothelium that is requisite for leukocyte capture via PSGL-1 in shear flow. Rather, this is a technical tour-de-force that reveals the complex dynamics following stimulation with histamine and its relationship to the molecular structure of P-selectin. Gene editing techniques were used to alter domains and to determine their relative contributions to the rate of diffusion and pause times following exocytosis. The authors quantified rapid diffusion of P-selectin from Weibel Palade bodies early following fusion that then appeared to slow down as they link to the cytoskeleton and with extracellular heparin sulfate of the glycoalyx. While the authors aptly provide evidence for the diffusive and pausing behaviors of P-selectin many questions arise involving the validation of their model with regards to more physiological scenarios of endothelial inflammation. P-selectin signaling characteristics, endocytosis characteristics, and their ultimate role in providing suitable mechanical anchors for leukocytes within a hemodynamic environment should be performed under inflammatory conditions. Additional validation would round out the main findings and produce a more impactful study.

Major Points:

1. While alteration of P-selectin dynamics in static and with histamine selectin provides a granular view of the diffusive characteristics of the molecule during exocytosis. It would be of critical importance to identify if the current times of diffusion are stimulus dependent. Suggest to characterize the diffusive behavior under additional stimulus, including IL-1 or TNF- α that are known to induce P-selectin exocytosis.
2. Additionally, it would be of interest to determine if diffusion characteristics of P-selectin are influenced by the presence of shear stress acting upon the EC monolayer (i.e. high vs low levels of shear stress). It is well established that shear stress superposes with activation via inflammatory stimuli. Does this also altered as a function of endothelial alignment? This will strengthen the model presented in Figure 3.
3. It would be of critical importance to demonstrate the impact of WT versus mutant and P-selectin diffusion on the capture of leukocytes under shear conditions. It is especially relevant to characterize the influence of tether formation on limiting ligated versus non-ligated P-selectin diffusion and pausing dynamics as this captures an environment with physiological drag forces and bond formation. This

would be especially critical in understanding subsequent adhesive events promoted by selectin upregulated on the endothelium.

4. While the authors mention P-selectin pausing and restriction later restricted to by heparin sulfate in later figures. Authors should demonstrate whether subcellular localization is dependent upon downstream cytokine signaling. Do these molecules locate near the Endoplasmic Reticulum which can undergo stress?

Minor Comments:

1. It is mentioned later in the paper, but not clearly explained how single P-selectin molecules are identified during exocytosis from WP bodies versus P-selectin aggregates.
2. For the figures, should use more distinguishable colors for the steady state condition and delimiting the different times of exocytosis and diffusion.
3. Figure 2 the construction and analysis of the P-selectin mutants and the impact of receptor diffusion is well constructed to rule out lipid mixing as an influence. While the role of the CTLD ectodomain is well defined in later figures, the independent effect of the CT domains remains ill-defined in terms of limiting diffusion/mediating pausing based on potential homophilic interactions or recruitment of P-selectin to cellular substructures.
4. The authors mention a potential role for AP2 at the level of the P-selectin CT in inducing diffusive/pausing events. Why not test the importance of the CT using the mutants with AP2 KD or overexpression to test this?
5. Additionally, the authors should apply more convention methods such as cytochalasin/actin polymerization inhibition to test active cytoskeletal remodeling in the process of P-selectin diffusion and localization.
6. Figure 4A's use of the antibodies provides a confirmation of the EGFP technique for tracking P-selectin; this can likely be moved to a supplementary figure.
7. The authors should comment on the function of the immobile clusters of P-selectin relative to the more diffusive or limited diffusion of the exocytosed selectin? It is currently difficult to parse the importance of this clustering behavior beyond a descriptive basis in the manuscript.
8. It is not clear how many individual experiments correlated to the number of cells analyzed. There should be separate experiments used to assess the main results with replicates included.

Reviewer #2 (Remarks to the Author):

Hellen et al use single molecule tracking approaches to describe dynamics of the leukocyte receptor P-selectin after exocytotic incorporation into the plasma membrane of endothelial cells. Exocytosed P-selectin exhibits a biphasic behavior characterized by a fast initial diffusion followed by a more restricted diffusion and the formation of P-selectin clusters that become apparent appr. 2 to 3 minutes after secretion. Recording the properties of P-selectin mutants revealed that both the extracellular C-type lectin domain and the cytoplasmic tail contribute to limiting P-selectin surface diffusion, most likely in an independent and additive manner. As heparinase treatment has similar effects, the authors conclude that interaction of the CTLD with extracellular glycans is involved in restricting free lateral diffusion.

This is a carefully performed study that shows changes in the surface movement/diffusion of P-selectin after exocytotic delivery of the protein from endothelial Weibel Palade bodies to the plasma membrane. The data deserve publishing, however they are limited in their degree of novelty and interest to others in the community. It is already known that P-selectin clusters form on the endothelial cell surface following exocytosis and that these clusters are relevant for mediating leukocyte contact/rolling. Moreover, it has been shown by Doyle et al (Blood 2011) that this clustering requires the presence of CD63, most likely forming tetraspanin enriched domains containing P-selectin. As a matter of fact these domains also preclude rapid re-internalization of P-selectin. The contribution of these domains (and CD63), the regulation of the switch between fast and restricted diffusion, the effect of P-selectin mutants on internalization and the role of the clusters described here in leukocyte rolling have all not been addressed.

Reviewer #3 (Remarks to the Author):

In this manuscript, Hellen et al. describes that P-selectin diffuses freely upon release from exocytosis sites, but changes to restricted diffusion after some minutes. The change to restricted diffusion (increase of immobile fraction) is shown to depend on both (1) the intracellular cytoplasmic domain and (2) the extracellular C-type lectin domain (tested with dCTL). The authors tested heparinase III to remove cell surface heparan sulfate (HS) which resulted in similar MSD plots compared to dCTL. This suggests that restricted diffusion is due to interactions with cell surface HS. Furthermore, the authors speculate that the CT domain might interact with clathrin coated pits also leading to reduced p-selectin mobility (tested via the dCT construct).

The paper is a systematic study that demonstrates how to extract biological information from single particle tracks (SPT).

Specific comments for revision:

MAJOR:

1. The higher the density of single molecules at a certain region of interest the higher the chance of false trajectory connections. The exocytosis event starts with a very high density and might continue to be very dense in the 0-40s time interval. What is the density of single P-selectins during the first 40s in the region of which you extract the trajectories (e.g. in a circle of $r=4\mu\text{m}$)? Could you do a simulation of the exocytosis event (using the experimental density and diffusion constants) on a pixelized camera to check if your extracted single molecule localizations and resulting trajectories are giving the right values.

Or alternatively could you bleach the pre-fusion exocytosis area to reduce the fluorescence density and check if you get the same msd-t plots.

2. You state that 'in time' ($>120\text{s}$) p-selectin gets immobile (120-180s msd-t plot decreases for long times). Could it be that it is not the time but the distance to the exocytosis point?

Is the msd plot of trajectories in a $3\mu\text{m}$ circle around the exocytosis midpoint compared to a ring ($5\mu\text{m}-10\mu\text{m}$) around the exocytosis midpoint the same?

(for $t_{\text{exo}} = 0-40\text{s}$ and $t_{\text{exo}} = 120-180\text{s}$)

Do you get the same msd plot for 20-40s compared to 0-20s?

3. Equation 1 allows to fit a restricted msd-t plot. You described in the text a variance-weighted least square minimization to solve the equations for 5 parameters. I could not find the extracted lumped stiffness and damping factors from the fit?

Can you comment on the goodness of such a fitting procedure.

4. You speculate that the cytoplasmic domain of p-selectin leads to mobility restrictions and clustering due to interactions with AP2 μ 2 and/or CCP. Could you show that indeed dCT is less colocalized with CCPs compared to wt.

5. Cytoplasmic domains are shown to interact or being confined within the actin cytoskeleton. Could you test if disruption of the actin meshwork (LatA, cytoD) leads to an increase in mobility but stays unaffected for the dCT construct.

Minor

1. How do you define the exact start point (the exocytosis event) in order to get the trajectory time frame 0-40s?
2. Do you allow multiple exocytosis events in the time frame 0-180s and/or steady state? And if so, how do you separate trajectories during multiple exocytosis events?
3. How many WBL bodies are undergoing exocytosis during time (exo/h) and how many single P-selectins proteins are typically in one exocytosis event?
4. You are referring to a sol-gel transition. Please specify more details. Do you see 'rafts' or (dis)-ordered domains or phase separations in the plasma membrane?

Do you foresee a relation to the stiffness or damping factors?

How could a single molecule immobilization event (binding to another protein) be related to a phase transition?

We thank the three referees for their careful and very helpful review of our earlier submission and we have now undertaken considerable new experimental work to address their concerns including: **1)** AP2 knock-down and overexpression with parallel controls for Δ CTLD and full-length P-selectin; **2)** cytochalasin D treatment with parallel controls; **3)** interleukin IL-1 β treatment; **4)** additional simulations to check if tracking of mobility is affected by changes in single fluorophore density at the different time points; **5)** confocal microscopy showing cross-correlation of eGFP- Δ CT and eGFP-WT with mCherry-clathrin; **6)** We also performed extensive time-lapse imaging of WT in an effort to address the excellent suggestion of Ref 3 to see if mobility correlated with distance moved from the site of exocytosis – these experiments basically failed due to technical limitations (as we discuss below). **7)** Finally, we have made changes to the manuscript text to address various point raised by the referees and have also made small editorial and typographical changes to maintain a consistent flow in the text.

Referees original comments are pasted in black, our responses are in blue and highlighting corresponds to substantive changes in the Manuscript, Figures, Methods and Supplementary Materials.

Reviewer #1 (Remarks to the Author):

The manuscript of Hellen et. al., examined the trajectory of P-selectin secreted from Weibel Palade bodies following histamine stimulation. This study lacked a stated hypothesis to guide the major question of how P-selectin diffuses across the plasma membrane of activated endothelium that is requisite for leukocyte capture via PSGL-1 in shear flow.

We draw the reviewer's attention to page 2 line 23ff

Importantly, we do not understand...

and Page 3 Lines 23ff

In the current study..

Rather, this is a technical tour-de-force that reveals the complex dynamics following stimulation with histamine and its relationship to the molecular structure of P-selectin.

We are pleased that Reviewer 1 acknowledges the technical challenges involved in dynamic imaging of single GFP fluorophores in live cells at 37oC. Which has enabled our new and important findings on time-dependent changes in P-selectin mobility that are critical to its biological function.

Gene editing techniques were used to alter domains and to determine their relative contributions to the rate of diffusion and pause times following exocytosis. The authors quantified rapid diffusion of P-selectin from Weibel Palade bodies early following fusion that then appeared to slow down as they link to the cytoskeleton and with extracellular heparin sulfate of the glycoalyx.

We did not claim that P-selectin binds the cytoskeleton and we have now specifically tested the effect of disrupting the actin cytoskeleton by cytochalasin D. It has no significant effect on anomalous diffusive behaviour at steady-state.

See page 7 In 15 ff

Identifying molecular interactions that limit P-selectin mobility: To further investigate....

While the authors aptly provide evidence for the diffusive and pausing behaviors of P-selectin many questions arise involving the validation of their model with regards to more physiological scenrios of endothelial inflammation. P-selectin signaling characteristics, endocytosis characteristics, and their ultimate role in providing suitable mechanical anchors for leukocytes within a

hemodynamic environment should be performed under inflammatory conditions. Additional validation would round out the main findings and produce a more impactful study.

Our study relates to stimulated secretion of P-selectin but we have now specifically tested the effect of pro-inflammatory IL-1 β stimulation as requested by the reviewer - it had no effect, see Major point 1 below and..

Page 8 Line 21:

Finally, to test whether P-selectin mobility is altered in cells exposed to pro-inflammatory stimuli...

Major Points:

1. While alteration of P-selectin dynamics in static and with histamine selectin provides a granular view of the diffusive characteristics of the molecule during exocytosis. It would be of critical importance to identify if the current times of diffusion are stimulus dependent. Suggest to characterize the diffusive behavior under additional stimulus, including IL-1 or TNF- α that are known to induce P-selectin exocytosis.

Thank you for this helpful suggestion. We have now undertaken new experiments and analysed steady-state diffusion of WT P-selectin-GFP in cells pre-treated with Interleukin-1 (1ng/ml, 24 h). The result showed no significant change in the diffusive properties of P-selectin under conditions where a range of inflammatory cytokines are upregulated in the cells. We show the activation of cells by IL-1 β and the distributions of lateral diffusion coefficients for WT P-selectin in treated and control cells in Supplemental Figure S12. We mention this finding in the text.

P8 ln 21 ff

Finally, to test whether P-selectin mobility is altered in cells exposed to pro-inflammatory stimuli, we treated HUVEC with IL-1 β ...

2. Additionally, it would be of interest to determine if diffusion characteristics of P-selectin are influenced by the presence of shear stress acting upon the EC monolayer (i.e. high vs low levels of shear stress). It is well established that shear stress superposes with activation via inflammatory stimuli. Does this also altered as a function of endothelial alignment? This will strengthen the model presented in Figure 3.

Although potentially interesting, this is totally beyond the scope of our paper and technically unfeasible for us – see response to major point 3 below.

3. It would be of critical importance to demonstrate the impact of WT versus mutant and P-selectin diffusion on the capture of leukocytes under shear conditions. It is especially relevant to characterize the influence of tether formation on limiting ligated versus non-ligated P-selectin diffusion and pausing dynamics as this captures an environment with physiological drag forces and bond formation. This would be especially critical in understanding subsequent adhesive events promoted by selectin upregulated on the endothelium.

These are interesting questions, some of which have been addressed previously by others; Roger McEvers group (Patel, K. D. Nollert, M. U.McEver, R. P. 1995, J Cell Biol, 131, 1893-) has previously looked at the effect of a range of P-selectin truncation mutants on leukocyte capture and rolling under flow. That study was in part the inspiration for this work where we have focused on how such mutations might affect the diffusive behaviour of P-selectin itself. To repeat previously reported cell adhesion experiments is beyond the scope of this study. To investigate the effect of fluid shear is technically beyond the scope of this study. The technique we use to detect and track the positions of single eGFP molecules (TIRF microscopy) is not well-suited for imaging single eGFP molecules on

the upper membrane of cells that would be exposed to fluid forces. Even with the kinds of modifications we made to image apical P-selectin (anti-P-sel-Ab-Cy3B) required the cells to be kept in a very stable and unperturbed environment (we use a sealed Rose chamber mounted on a very stable stage) in order to minimise cell vibrations or micro movements that hamper detection and tracking of mobile SFs. Although very interesting we are currently unable to image P-selectin tether formation and the effect of physiological drag forces and bond formation at a single molecule level on the apical surface of these cells.

4. While the authors mention P-selectin pausing and restriction later restricted to by heparin sulfate in later figures. Authors should demonstrate whether subcellular localization is dependent upon downstream cytokine signaling. Do these molecules locate near the Endoplasmic Reticulum which can undergo stress?

This is an interesting point. We have shown, using standard-deviation projections of TIRFM image sequences, the localisation of proteins to reticular ER structures within cells (e.g. Mashanov *et al.* 2021, Faraday Discussions, **232**:358 - 374) and applied this approach to HUVEC expressing WT P-selectin-eGFP under control or IL-1 β treated conditions. In neither case did we see any reticular pattern that might indicate association with ER elements close to the plasma membrane.

Minor Comments:

1. It is mentioned later in the paper, but not clearly explained how single P-selectin molecules are identified during exocytosis from WP bodies versus P-selectin aggregates.

The following text was added to explain our criteria of single molecule identification:

page 4, line 3 ff

Single molecules (SMs) were identified by their characteristic.....

2. For the figures, should use more distinguishable colors for the steady state condition and delimiting the different times of exocytosis and diffusion.

We have changed the color-coding for all Figure plots to make them more easily distinguishable.

3. Figure 2 the construction and analysis of the P-selectin mutants and the impact of receptor diffusion is well constructed to rule out lipid mixing as an influence. While the role of the CTLD ectodomain is well defined in later figures, the independent effect of the CT domains remains ill-defined in terms of limiting diffusion/mediating pausing based on potential homophilic interactions or recruitment of P-selectin to cellular substructures.

4. The authors mention a potential role for AP2 at the level of the P-selectin CT in inducing diffusive/pausing events. Why not test the importance of the CT using the mutants with AP2 KD or overexpression to test this?

Response to 3&4: Thank you for suggesting we look more closely at the contribution of AP2. We have now undertaken an extensive series of new experiments to investigate the role of AP2 in shaping P-selectin mobility by RNAi mediated disruption of the AP2 complex and by over-expression of the Ap2m2 subunit (the P-selectin binding subunit of AP2). We found that after AP2 disruption the MF and limiting MSD for WT P-selectin were significantly increased, such that WT now resembled more closely the behaviour of the delta CT construct. A similar change was seen with the delta-CTLD mutant. Over-expression of the AP2 μ 2 subunit had the opposite effect (as would be expected if CT-AP2 interactions inhibit mobility). Together the new data provide evidence that interaction between P-selectin and AP2 plays an important role in shaping the mobility at steady-state. The new data substantially enhances our study by providing a more detailed mechanistic insight into the factors that impact on P-selectin behaviour at the plasma membrane. **Please see our new Figure 5 and supporting data and Figures S8, S9, S10 & S11.**

See new text on P.7 In 15ff

Identifying molecular interactions that limit P-selectin mobility: To....

5. Additionally, the authors should apply more convention methods such as cytochalasin/actin polymerization inhibition to test active cytoskeletal remodeling in the process of P-selectin diffusion and localization.

Thank you for this suggestion. We have now tested the effect of cytochalasin D mediated disruption to the actin cytoskeleton on WT P-selectin diffusion and found that it has remarkably little effect (Figure S7).

We have added text on P7 In 15:

To further investigate the molecular basis for the independent contribution of the CT to P-selectin mobility we first disrupted the actin cytoskeleton using cytochalasin D....

6. Figure 4A's use of the antibodies provides a confirmation of the EGFP technique for tracking P-selectin; this can likely be moved to a supplementary figure.

We would prefer to keep Figure 4a in the main text as it shows examples of the pauses in mobility identified by using the Cy3B labelled P-selectin molecules. The controls for these experiments are detailed in the Supplementary data Figures S4,5.

7. The authors should comment on the function of the immobile clusters of P-selectin relative to the more diffusive or limited diffusion of the exocytosed selectin? It is currently difficult to parse the importance of this clustering behavior beyond a descriptive basis in the manuscript.

We have added text in the introductory section to signpost the relevance of clustering.

P3 In 18ff

Clustering of P-selectin is seen at later times...

..and further text in the results section.

P9 In 2ff

At steady-state we observed P-selectin-eGFP clustering into immobile puncta

8. It is not clear how many individual experiments correlated to the number of cells analyzed. There should be separate experiments used to assess the main results with replicates included.

See Methods section:

P13 In 28ff

Statistical analysis. We carried out between 3 and 9 replicate experiments for WT P-selectin and each of the..

Reviewer #2 (Remarks to the Author):

Hellen et al use single molecule tracking approaches to describe dynamics of the leukocyte receptor P-selectin after exocytotic incorporation into the plasma membrane of endothelial cells. Exocytosed P-selectin exhibits a biphasic behavior characterized by a fast initial diffusion followed by a more

restricted diffusion and the formation of P-selectin clusters that become apparent approx. 2 to 3 minutes after secretion. Recording the properties of P-selectin mutants revealed that both the extracellular C-type lectin domain and the cytoplasmic tail contribute to limiting P-selectin surface diffusion, most likely in an independent and additive manner. As heparinase treatment has similar effects, the authors conclude that interaction of the CTLD with extracellular glycans is involved in restricting free lateral diffusion.

This is a carefully performed study that shows changes in the surface movement/diffusion of P-selectin after exocytotic delivery of the protein from endothelial Weibel Palade bodies to the plasma membrane. The data deserve publishing, however they are limited in their degree of novelty and interest to others in the community. It is already known that P-selectin clusters form on the endothelial cell surface following exocytosis and that these clusters are relevant for mediating leukocyte contact/rolling. Moreover, it has been shown by Doyle et al (Blood 2011) that this clustering requires the presence of CD63, most likely forming tetraspanin enriched domains containing P-selectin. As a matter of fact these domains also preclude rapid re-internalization of P-selectin. The contribution of these domains (and CD63), the regulation of the switch between fast and restricted diffusion, the effect of P-selectin mutants on internalization and the role of the clusters described here in leukocyte rolling have all not been addressed.

We thank the reviewer for their positive comments on the care and rigour of our study. Our objective was to investigate how the structure of P-selectin might influence its mobility in the plasma membrane after secretion. We have shown directly for the first time how P-selectin spreads from exocytosis sites and changes its behaviour in time after secretion. We have identified interactions of specific structural elements within the extracellular domain of P-selectin and cell surface components that shape that behaviour, and our new experiments have shown that AP2, the best-characterised cytoplasmic interactor of the P-selectin cytoplasmic tail is also responsible for shaping the steady-state behaviour of P-selectin. We describe these new findings on:

Page 7 Line 15ff

Identifying molecular...

The idea that CD63 plays some role in P-selectin clustering and how this might impact on the transitions in mobility we see after secretion, is very interesting and something we are beginning to look at in more detail. Our preliminary single WPB FRAP analysis so far indicates that the presence of P-selectin does not change the mobility of CD63 within the limiting membranes of individual WPBs, and preliminary SF analysis of P-selectin and CD63 co-secretion suggest they diffuse independently at and after the point of secretion. However, to properly address the role of CD63 and clusters is going to be a complex and long-term project that is simply beyond the resources and scope of this study. We hope the reviewer understands.

Reviewer #3 (Remarks to the Author):

In this manuscript, Hellen et al. describes that P-selectin diffuses freely upon release from exocytosis sites, but changes to restricted diffusion after some minutes. The change to restricted diffusion (increase of immobile fraction) is shown to depend on both (1) the intracellular cytoplasmic domain and (2) the extracellular C-type lectin domain (tested with dCTLCD). The authors tested heparinase III to remove cell surface heparan sulfate (HS) which resulted in similar msd plots compared to dCTLCD. This suggests that restricted diffusion is due to interactions with cell surface HS. Furthermore, the authors speculate that the CT domain might interact with clathrin coated pits also leading to reduced p-selectin mobility (tested via the dCT construct).

The paper is a systematic study that demonstrates how to extract biological information from single particle tracks (SPT).

Specific comments for revision:

MAJOR:

1. The higher the density of single molecules at a certain region of interest the higher the chance of false trajectory connections. The exocytosis event starts with a very high density and might continue to be very dense in the 0-40s time interval. What is the density of single P-selectins during the first 40s in the region of which you extract the trajectories (e.g. in a circle of $r=4\mu\text{m}$)? Could you do a simulation of the exocytosis event (using the experimental density and diffusion constants) on a pixelized camera to check if your extracted single molecule localizations and resulting trajectories are giving the right values.

Or alternatively could you bleach the pre-fusion exocytosis area to reduce the fluorescence density and check if you get the same msd-t plots.

The reviewer is absolutely correct to point out difficulties in tracking single GFP fluorophores in live cells when they are present at high surface densities. We have tested and validated our imaging and tracking approaches in earlier publications and we applied the same approaches to the current data sets (see Supplementary Materials of Nenasheva *et al.* (2013) *J. Mol. Cell. Cardiol.* **57**:129–136).

To validate our single fluorophore tracking methodology in the current work, we used our Monte Carlo simulation software (Ref 24 main text: Mashanov (2014) *J. Roy. Soc. Interface* **11**:20140442) to generate mock or “fake” video data sets that have identical signal-to-noise ratio, photobleaching rate, in order to mimic our experimental data sets.

See P4 In 3ff

Single molecules (SMs) were identified by their characteristic diffraction-limited spot-size, single GFP intensity level and single-step photobleaching behaviour²². SMs were localized with 25 nm precision in each video frame and tracked from frame-to-frame using image analysis software²³ (Figure 1a-d). We validated our ability to track individual molecules present at different surface densities and with different diffusion coefficients using simulated data sets with identical signal-to-noise to our real data sets as described previously²⁴.

In our simulations, we systematically vary the density of fluorophores diffusion rates and type of diffusive process (random, motorised or anomalous diffusion) to test the robustness of our analysis tools. The fluorophore density, at the time of release, (see Video S1) is high and tracking does not work well until fluorophore density falls due to diffusion (i.e. fluorophores spreading out across the membrane) and photobleaching over the first few seconds following fusion and release. The differences we see between full-length P-selectin and the different mutants are internally controlled because the exocytotic events release similar numbers of molecules.

2. You state that ‘in time’ (>120s) p-selectin gets immobile (120-180s msd-t plot decreases for long times). Could it be that it is not the time but the distance to the exocytosis point?

Is the msd plot of trajectories in a $3\mu\text{m}$ circle around the exocytosis midpoint compared to a ring ($5\mu\text{m}$ - $10\mu\text{m}$) around the exocytosis midpoint the same?

(for $t_{\text{exo}} = 0-40\text{s}$ and $t_{\text{exo}} = 120-180\text{s}$)

Do you get the same msd plot for 20-40s compared to 0-20s?

The reviewer proposes a really neat experiment - which is to analyse our data as an “Archery Target” of concentric rings extending out from a fusion site and report the diffusive behaviour at different distances from the fusion origin at different times. By doing this we might then be able to distinguish whether our observed mobility changes are a function of distance (from the fusion site) or time after fusion. Of course, distance and time are linked but the random nature of diffusion would in principle allow this idea to be tested.

To test the review’s idea we performed a series of experiments using time-lapse imaging (taking 1 second bursts of video leaving 15 seconds (with laser excitation gated off) between data capture intervals. However, we were hampered with two different problems:

1) Multiple fusion events nearly always occurred in the same cell following stimulation and it was impossible to measure off distance in a reliable way from any one particular fusion site (the “archery targets” overlapped to such an extent that we could not produce meaningful distance maps)

2) MSD vs dT analysis requires a large number of molecular trajectories (many hundreds) to produce a reliable plot and it was difficult to produce reliable plots once our data sets had been “decimated” on the basis of linear distance from the site of fusion. In fact, we have performed very similar experiments in order to study spatial variations in lateral diffusion using G-protein coupled receptors labelled with cy3B fluorophores (Mashanov *et al.* (2021) *Faraday Discussions* **232**:358 – 374) That study highlights the statistical problems in producing high-resolution diffusion maps. Doing this with eGFP fluorophores is really very demanding – and to cut a long story short, in this instance, we tried hard but sadly failed.

3. Equation 1 allows to fit a restricted msd-t plot. You described in the text a variance-weighted least square minimization to solve the equations for 5 parameters. I could not find the extracted lumped stiffness and damping factors from the fit?

We now state the fitting parameters obtained by global fitting in the Figure Legend and Main text
page 6 ln 8ff

Global fitting (Microsoft Excel “Solver” function) to all of the MSD vs dT data sets by variance-weighted...

Can you comment on the goodness of such a fitting procedure.

The goodness to fit is stated as 95% confidence intervals (shown in the histograms of Figures 1, 2, and 3). This is now stated in the figure legends.

4. You speculate that the cytoplasmic domain of p-selectin leads to mobility restrictions and clustering due to interactions with AP2 μ 2 and/or CCP. Could you show that indeed dCT is less colocalized with CCPs compared to wt.

We performed dual-colour confocal microscopy on co-labelled AP2 μ 2-mCherry with full-length and Δ CT eGFP-P-selectin and show the expected change in co-localisation (Figure S11)

P8 ln 4ff.

We next over-expressed AP2 μ 2-mCherry and confirmed that it co-localized with the endogenous AP2 complex and P-selectin-eGFP in a CT-dependent fashion (Figure S11)...

5. Cytoplasmic domains are shown to interact or being confined within the actin cytoskeleton. Could you test if disruption of the actin meshwork (LatA, cytoD) leads to an increase in mobility but stays unaffected for the dCT construct.

We performed additional work and disrupted the actin cytoskeleton using cytochalasin-D (Figure S7) and show that mobility is NOT affected by the actin cytoskeleton (as has been shown by other workers (Ref 31 (and refs within) and our own work studies (Ref 45) for other proteins. This is consistent with the small size of the CT region and the fact that it has been shown in previous studies not to interact with actin directly (Ref 20).

P7 ln 15ff

Identifying molecular interactions that limit P-selectin mobility: To further investigate the molecular basis for the independent...

Minor

1. How do you define the exact start point (the exocytosis event) in order to get the trajectory time frame 0-40s?

The video recording is started before addition of histamine and there is usually a delay of a few seconds before the onset of exocytotic events. We define the exocytosis start-time “t=0” individually for each event as the moment when WPB fusion occurs – which is easily visible in the records (see Video S1)

Figure 1 Legend: Page 19 In 6

...B) 40 s period after exocytosis (“0-40 s”) triggered by 100 μ M histamine; “time zero” is defined as the moment that fusion occurs.

2. Do you allow multiple exocytosis events in the time frame 0-180s and/or steady state?

Yes and No: Cells are selected before stimulation on the basis of being well-spread, “healthy looking”, with visible WPB trafficking and a suitable density of well-spaced WPBs present (usually around 10-20 visible WPB’s per cell). Following stimulation and recording from a single cell, the entire coverslip is discarded and replaced so that another, unstimulated, cell on a new coverslip can be selected and visualised. So, the work is technically challenging and labour-intensive with only one round of stimulation per experiment.

Following histamine stimulation, multiple exocytosis events (1 to 20) occur within the same cell and at approximately the same time (+/-5 seconds). The early recordings “0-40s” are individually timed and synchronised so each fusion event analysed has its own start time “t=0”. The later data collection time window “120-180s” will indeed have some degree of timing “jitter” since molecules released from different fusion sites will have spread out and will overlap. The time is relative to the first fusion event observed within that cell. The dispersion of start times (“t=0”) is small relative to the time window.

And if so, how do you separate trajectories during multiple exocytosis events?

Because exocytosis sites are usually >5 μ m apart, after ~40 seconds it becomes impossible to tell which molecule came from which exocytosis site. For later time windows (>40 seconds) it is impossible to know from where the molecules arose because laser excitation is “gated off” from “t=0” until the desired collection time-window to prevent photobleaching.

3. How many WBL bodies are undergoing exocytosis during time (exo/h) and how many single P-selectins proteins are typically in one exocytosis event?

Typically, each exocytosis event releases on the order of 100 eGFP-tagged-P-selectin molecules (based on the integral intensity at t=0). Only a subset of them (usually <50 molecules) can be reliably tracked over the different windows studied. However, because tracking sometimes fails (resulting in a “broken track”) each exocytosis typically gives rise to 30-60 trajectories (each track lasting >15 video frames (>0.5 s)). However, some events can give rise to fewer tracks (10->20) because of image quality and other technical issues.

The detail to questions 2&3 above will be addressed in our Supplementary Data spreadsheets which will be provided with the manuscript according to *Nature* Portfolio reporting requirements.

4. You are referring to a sol-gel transition. Please specify more details. Do you see ‘rafts’ or (dis)-ordered domains or phase separations in the plasma membrane?

We do not think the plasma membrane lipids undergo a phase-transition, in fact, the linear MSD vs dT plots exhibited by our TMD mutant indicate the endothelial cell membrane remains fluid and isotropic. Rather, it appears that full-length P-selectin first moves freely within the lipid bi-layer but then, over time, it becomes “trapped” or “ensnared” in a gel-like phase and it is interactions mediated

via CT with AP2 and CTLD with heparan sulphate which restrict its motion, so it appears to become “set” in a gel.

Do you foresee a relation to the stiffness or damping factors?

The damping factors make good sense and are easy to understand because they match the expected lateral diffusion coefficient (within a factor of 2 or so) for transmembrane proteins ($D \sim k_B T / \beta$). The “stiffness” parameter is less easily understood, but may relate to the deformability of macromolecular networks or the lifetime of binding interactions. A future line of work might be to use optical tweezers or some other physical method to drag P-selectin through the membrane and measure β and κ terms directly. But that is beyond the scope of the current study.

How could a single molecule immobilization event (binding to another protein) be related to a phase transition?

The simple answer would be to say that the binding event rate must change. Immediately following exocytosis P-selectin, for some reason, does not interact much with the extra- and intra-cellular molecules but those interactions then occur more frequently as time passes. We really do not understand the time-dependence of these processes – it seems that “fresh” P-selectins diffuse relatively fast and freely whereas “old” P-selectins move in an anomalous fashion. We don’t yet understand the chemistry but we do know that the CT and CTLD regions are important and that the effects are in parallel and additive.

REVIEWER COMMENTS

Reviewer #1 (Remarks to the Author):

The resubmission by Hellen et. al. uses TIRF and single molecule tracking techniques to characterize the dynamics of P-selectin diffusion and immobilization following histamine-stimulated release from WPBs in HUVEC. The study investigates a molecular basis for the time-dependent changes in P-selectin mobility, which likely have important ramifications related to its role in leukocyte capture- although such studies were not examined.

The study is carefully executed and technically sound. The authors have been generally responsive to the previous review and have included new data that strengthen the conclusions regarding the molecule basis for the observed phenomenon.

The study might be further enhanced by the following:

1. In the rebuttal the authors lay out the case for not repeating studies involving shear stress or leukocyte capture. It is not clear that the cited studies have specifically characterized the impact of the deletion mutations used in the current study with respect to the effect on leukocyte rolling/ capture under flow. Minimally, previous work citing a specific effect of the CT and CTLD mutations on the number of adhering leukocytes on stimulated EC would serve to elucidate an outcome of functional importance to the observed phenomenon. The physiological relevance of the findings on leukocyte tethering should be included even if it is not directly linked to the single molecule tracking of the current study.
2. The authors indicate that the diffusion behavior can arise from different phenomena, including heterogeneity in lipid composition (e.g. lipid rafts). An experiment disrupting lipid rafts could demonstrate that this does not contribute to the observed behavior.
3. In Figure 6, the specificity of the role of the heparin sulfate – CTLD interaction in contributing to the behavior would be strengthened by including the data showing that removal of chondroitin sulfate and sialic acid did not affect the mobility.

Reviewer #2 (Remarks to the Author):

In this revised version the authors have included additional experimental data, in particular addressing the effect of endothelial stimulation on P-selectin mobility and potential interactions of the actin cytoskeleton and the clathrin adapter machinery with the C-terminal cytoplasmic tail of P-selectin that could explain the altered behaviour of the C-terminal tail deletion mutant. In response to my comment

concerning the potential role of CD63 in regulating P-selectin mobility, spreading and clustering after exocytosis the authors mention preliminary experiments suggesting that P-selectin and CD63 diffuse independently after secretion. These experiments are however not shown. They would argue against a role of CD63 in stabilizing P-selectin clusters and thus would represent an important novel result, in particular in light of the published findings that CD63-deficient endothelial cells (HUVEC) fail to recruit leukocytes and CD63 KO mice show defects in (P-selectin mediated) leukocyte rolling, adhesion and subsequent extravasation in inflammatory scenarios (Doyle et al., Blood 118, 4265-4273, 2011).

Reviewer #3 (Remarks to the Author):

The Authors have carefully addressed my questions.

Specifically they added details about tracking, carried out simulations of the process and tried various additional data analysis methods.

From my side the data, conclusions, interpretations and claims deserve publishing.

We thank the referees for their further comments and helpful review of our resubmitted manuscript. We have undertaken some further analysis and made suitable amendments to the text and we have produced a revised **Figure 6** that now shows all of our data investigating the effects of neuraminidase and chondroitinase compared to heparinase III treatment. Reviewer 1 also requested a totally new study to investigate the effects of removing cholesterol from the plasma membrane: This is beyond the scope of our current paper.

We include results from our preliminary P-selectin::CD63 single fluorophore co-localisation experiments performed in live-cells which we report in Supplementary Materials (**Fig. S12**). The work is at an early stage and has shortcomings, as we acknowledge in the text. We are happy to share our results so that specialist readers can explore the preliminary data relating to the point made by Reviewer 2.

The Reviewers' original comments are pasted in black, **our responses are in blue** and **highlighting** corresponds to changes in the Manuscript and Supplementary Materials and on-line source data file..

Reviewer #1 (Remarks to the Author):

The resubmission by Hellen et. al. uses TIRF and single molecule tracking techniques to characterize the dynamics of P-selectin diffusion and immobilization following histamine-stimulated release from WPBs in HUVEC. The study investigates a molecular basis for the time-dependent changes in P-selectin mobility, which likely have important ramifications related to its role in leukocyte capture- although such studies were not examined.

The study is carefully executed and technically sound. The authors have been generally responsive to the previous review and have included new data that strengthen the conclusions regarding the molecule basis for the observed phenomenon.

The study might be further enhanced by the following:

1. In the rebuttal the authors lay out the case for not repeating studies involving shear stress or leukocyte capture. It is not clear that the cited studies have specifically characterized the impact of the deletion mutations used in the current study with respect to the effect on leukocyte rolling/capture under flow. Minimally, previous work citing a specific effect of the CT and CTLD mutations on the number of adhering leukocytes on stimulated EC would serve to elucidate an outcome of functional importance to the observed phenomenon. The physiological relevance of the findings on leukocyte tethering should be included even if it is not directly linked to the single molecule tracking of the current study.

The reviewer is correct that this paper does not address interactions between the CTLD and its cognate leukocyte receptor, PSGL-1. These interactions have been extensively studied by other groups over many years and there would be little value in demonstrating that removal of the CTLD region results in loss of leukocyte capture and rolling. It has been established beyond reasonable doubt that the CTLD region binds PSGL-1, not least by the published crystal structure of the CTLD-PSGL1 complex.

We highlight sentences in the text to help the reviewer understand the physiological importance of our work. We feel that we have adequately explained how our discovery of time-dependent changes in mobility of P-selectin allow it to spread rapidly across the plasma membrane then, through interactions mediated by its extra- and intra-cellular regions acting in parallel, it becomes progressively immobilised, making it better-suited to leukocyte capture.

2. The authors indicate that the diffusion behavior can arise from different phenomena, including heterogeneity in lipid composition (e.g. lipid rafts). An experiment disrupting lipid rafts could demonstrate that this does not contribute to the observed behavior.

This is an interesting new idea, but, in our paper we show that the isolated membrane-spanning region of P-selectin, “TMD”, moves in an unrestricted manner across the membrane at all times. The TMD would be the best construct to test for heterogeneity in membrane lipid-bilayer structures (e.g. rafts). But, since we don’t see anomalous diffusion of TMD in wildtype HUVECs in the first place, there is really nothing to test by conducting further cholesterol depletion experiments.

3. In Figure 6, the specificity of the role of the heparin sulfate – CTLD interaction in contributing to the behavior would be strengthened by including the data showing that removal of chondroitin sulfate and sialic acid did not affect the mobility.

This is an excellent suggestion and we have completely redrawn Figure 6 to include data from all experiments as requested so the results can be compared at a glance.

Figure 6 now shows all the enzymatic treatments.

Reviewer #2 (Remarks to the Author):

In this revised version the authors have included additional experimental data, in particular addressing the effect of endothelial stimulation on P-selectin mobility and potential interactions of the actin cytoskeleton and the clathrin adapter machinery with the C-terminal cytoplasmic tail of P-selectin that could explain the altered behaviour of the C-terminal tail deletion mutant. In response to my comment concerning the potential role of CD63 in regulating P-selectin mobility, spreading and clustering after exocytosis the authors mention preliminary experiments suggesting that P-selectin and CD63 diffuse independently after secretion.

These experiments are however not shown. They would argue against a role of CD63 in stabilizing P-selectin clusters and thus would represent an important novel result, in particular in light of the published findings that CD63-deficient endothelial cells (HUVEC) fail to recruit leukocytes and CD63 KO mice show defects in (P-selectin mediated) leukocyte rolling, adhesion and subsequent extravasation in inflammatory scenarios (Doyle *et al.*, Blood 118, 4265-4273, 2011).

We do not think that CD63 is unimportant, in fact far from it, and we cite Doyle *et al.* and have added an additional reference to the work of Toothill *et al.* (new ref 29) to acknowledge those important earlier studies. The mechanism by which CD63 augments P-selectin function in leukocyte adhesion remains unclear but may arise through the formation of CD63:P-selectin clusters. We present our preliminary data in Supplementary Fig. S12 and the on-line Source Data File. It shows that while most CD63 molecules diffuse independently of P-selectin we observed a small proportion, < 1% of coincident single molecule diffusing tracks, and a similar value derived by cross-correlation analysis. We also report a larger pool of immobile P-selectin:CD63 clusters (about 5 % of total tracks and a similar estimate obtained by cross-correlation analysis). It is important to note that our current findings are confounded by the problem of unlabelled P-selectin and CD63 which will be present (yet invisible) in our fluorescence tracking experiments. To drill down into this important question would require use of knock-out cell-lines and is beyond the scope of the current paper. In our revised

manuscript we have toned-down our discussion of CD63:P-selectin interactions and we now present our preliminary data and explain the important caveats in its interpretation in the Supplementary Fig. S12 so that readers can evaluate this aspect of the work if they so wish.

Reviewer #3 (Remarks to the Author):

The Authors have carefully addressed my questions.

Specifically, they added details about tracking, carried out simulations of the process and tried various additional data analysis methods.

From my side the data, conclusions, interpretations and claims deserve publishing.

REVIEWERS' COMMENTS

Reviewer #1 (Remarks to the Author):

Although it would be more impactful to demonstrate the importance of the P-selectin mutants to capture and rolling of leukocytes, We appreciate this is beyond the scope of the current work.

Reviewer #2 (Remarks to the Author):

The authors addressed my concern about the potential role of CD63 in regulating/restricting P-selectin mobility on the endothelial cell surface and now show their preliminary experiments analyzing the colocalization of P-selectin and CD63 in live HUVEC (Figure S12). They show a very limited colocalization of GFP-CD63 and antibody labeled P-selectin suggesting that the 'vast majority of P-selectin and CD63 moved freely and separately at the membrane' – although some static clusters containing both molecules were also observed. The caveat is of course, and the authors mention this, that the endogenous pool of CD63 and P-selectin (not visible in the tracking approach) will compete with the labeled molecules. Thus, as the authors correctly state, their number/fraction of coclusters containing CD63 and P-selectin must be considered lower bounds. Nonetheless, it is difficult to reconcile these very low numbers with the abundant coclustering of (endogenous) CD63 and P-selectin reported by Doyle et al (Blood 118, 4265-4273, 2011) using both, EM and proximity ligation of HUVEC fixed after stimulating WPB exocytosis. A relatively easy experiment to address the issue would be an analysis of the P-selectin behaviour in a CD63 knockdown background.

REVIEWERS' COMMENTS

Reviewer #1 (Remarks to the Author):

Although it would be more impactful to demonstrate the importance of the P-selectin mutants to capture and rolling of leukocytes, We appreciate this is beyond the scope of the current work.

We agree with the referee that in an ideal world one would like to know the full effect of all mutations on leukocyte capture and rolling. To do this properly would involve a large amount of experimental work and we would need to develop and fully-characterise a totally new assay system.

Reviewer #2 (Remarks to the Author):

The authors addressed my concern about the potential role of CD63 in regulating/restricting P-selectin mobility on the endothelial cell surface and now show their preliminary experiments analyzing the colocalization of P-selectin and CD63 in live HUVEC (Figure S12). They show a very limited colocalization of GFP-CD63 and antibody labeled P-selectin suggesting that the 'vast majority of P-selectin and CD63 moved freely and separately at the membrane' – although some static clusters containing both molecules were also observed. The caveat is of course, and the authors mention this, that the endogenous pool of CD63 and P-selectin (not visible in the tracking approach) will compete with the labeled molecules. Thus, as the authors correctly state, their number/fraction of coclusters containing CD63 and P-selectin must be considered lower bounds. Nonetheless, it is difficult to reconcile these very low numbers with the abundant coclustering of (endogenous) CD63 and P-selectin reported by Doyle et al (Blood 118, 4265-4273, 2011) using both, EM and proximity ligation of HUVEC fixed after stimulating WPB exocytosis. A relatively easy experiment to address the issue would be an analysis of the P-selectin behaviour in a CD63 knockdown background.

In broad terms, there is fair agreement between our observations and those of Doyle *et al.* 2011. We both observe co-clusters of multiple P-selectin and CD63 molecules on the plasma membrane of HUVEC. However, methodological differences between the two studies render quantitative comparison between the frequency of co-clusters difficult. Our analysis is geared specifically to single molecule detection in live cells which excludes some multi-molecule clusters, although these can be clearly seen on the plasma membrane. Doyle *et al* did not use single molecule approaches, and analyzed co-localization of multi-molecule clusters in fixed cells using indirect means. The discrepancy between the numbers of clusters observed by PLA and by EM analysis within the Doyle *et al.* study serves to show how difficult it is to make quantitative comparisons of co-clustering from data obtained using different methodological approaches. It is not easy to do knock-down experiments to publication standard without performing a large number of controls. This would

involve months (or more) of additional experimental work and give minimal additional insight into this important problem. It is really a major study in itself.